# A phase 1/2 clinical trial of invariant natural killer T cell therapy in moderate-severe acute respiratory distress syndrome

Terese C. Hammond[1,2,11], Marco A. Purbhoo[3,11], Sapana Kadel[3], Jerome Ritz [4], Sarah Nikiforow[4], Heather Daley[4], Kit Shaw[4], Koen van Besien[5], Alexandra Gomez-Arteaga[6], Don Stevens[7], Waldo Ortuzar[8], Xavier Michelet[3], Rachel Smith[3], Darrian Moskowitz[3], Reed Masakayan[3], Burcu Yigit[3], Shannon Boi [3], Kah Teong Soh [8], John Chamberland[3,8], Xin Song[8], Yu Qin[3,8], Ilya Mishchenko[8], Maurice Kirby[8], Valeriia Nasonenko[8], Alexa Buffa [3,8], Jennifer S. Buell[3], Dhan Chand[8], Marc van Dijk [3], Justin Stebbing [9,12] ✉ & Mark A. Exley [10,12]

Invariant natural killer T (iNKT) cells, a unique T cell population, lend themselves for use as adoptive therapy due to diverse roles in orchestrating immune responses. Originally developed for use in cancer, agenT-797 is a donor-unrestricted allogeneic ex vivo expanded iNKT cell therapy. We conducted an open-label study in virally induced acute respiratory distress syndrome (ARDS) caused by the severe acute respiratory syndrome-2 virus (trial registration NCT04582201). Here we show that agenT-797 rescues exhausted T cells and rapidly activates both innate and adaptive immunity. In 21 ventilated patients including 5 individuals receiving veno-venous extracorporeal membrane oxygenation (VV-ECMO), there are no dose-limiting toxicities. We observe an anti-inflammatory systemic cytokine response and infused iNKT cells are persistent during follow-up, inducing only transient donor-specific antibodies. Clinical signals of associated survival and prevention of secondary infections are evident. Cellular therapy using off-the-shelf iNKT cells is safe, can be rapidly scaled and is associated with an anti-inflammatory response. The safety and therapeutic potential of iNKT cells across diseases including infections and cancer, warrants randomized-controlled trials.

Invariant Natural Killer T cells (iNKT) cells are capable of discriminating normal from abnormal cells with iNKT deficiency resulting in susceptibility to infections and cancer[1–3]. They are attractive for use as adoptive cellular therapies, in part for their ability to identify and kill abnormal cells, thus augmenting protective immunity[1–3] (Supplementary Fig. 1). Because of their therapeutic potential, iNKT cells are being developed as off-the-shelf cell therapies, most notably against hematologic and solid malignancies, although anti-pathogen and anti-inflammatory roles have been described[1–7].

[1]Pulmonary Critical Care Sleep Medicine, Providence Saint John's Health Center, Santa Monica, CA, USA. [2]David Geffen School of Medicine at UCLA, Los Angeles, CA, USA. [3]MiNK Therapeutics, Lexington, MA, USA. [4]Dana Farber Cancer Institute, Boston, MA, USA. [5]UH Seidman Cancer Center, Cleveland, OH, USA. [6]Weill Cornell Medicine, New York, NY, USA. [7]Norton Cancer Center, Louisville, KY, USA. [8]Agenus, Lexington, MA, USA. [9]Anglia Ruskin University, Cambridge, UK. [10]Brigham & Women's Hospital, Boston, USA. [11]These authors contributed equally: Terese C. Hammond, Marco A. Purbhoo. [12]These authors jointly supervised this work: Justin Stebbing, Mark A. Exley. ✉e-mail: justin.stebbing@aru.ac.uk

Despite a very low frequency in human peripheral blood measuring 0.01–0.1%, iNKT cells are potent immune stimulators and possess both innate and adaptive immune cell surface markers. In particular, they express a highly restricted invariant T cell receptor (iTCR), recognizing glycolipid antigens presented by the non-classical MHC molecule, CD1d with stereo specificity[1–3]. A number of studies have delineated the importance of iNKT cells in coordinating both the innate and adaptive responses during bacterial and viral infections, and they also appear capable of both promoting cell-mediated immunity to tumors, as well as suppressing deleterious cell-mediated immunity associated with autoimmune disease and allograft rejection[8–16].

During viral infections, iNKT cell numbers are decreased rapidly in patients, for example in those with HIV after seroconversion[16]. Intriguingly, they have been shown to regulate airway hyper-reactivity and the iNKT CD1d-binding ligand, alpha-galactosylceramide (α-GalCer) potentiates immune responses to models of H1N1 influenza, viral encephalomyocarditis and also in HIV vaccine models[9,13–19]. While glycolipid-mediated stimulation of iNKT cells or their use as cell-based immunotherapy is already under clinical investigation for cancer[2–6,10] and α-GalCer has been used clinically in chronic viral infections[10], the antiviral activities of iNKT cells make them potentially suitable as an adoptive immunotherapy during acute viral infections, including sequelae of SARS-CoV-2[4,7,17–19].

To study their role further in humans, we have developed an ex vivo expanded and scalable allogeneic iNKT cell product (agenT-797) for treatment of a broad spectrum of diseases, including viral infections.

Here we show in a phase 1/2 study the feasibility of using an unmodified, allogeneic iNKT therapy, in participants with moderate-to-severe acute respiratory distress syndrome (ARDS) secondary to SARS-CoV-2 including critically unwell individuals receiving VV-ECMO. This is rapidly scalable with preliminary evidence of therapeutic efficacy, and we detect no adverse safety signals.

## Results
### Trial endpoints
Alongside the role of NK cells during viral infections and their therapeutic potential[20], we describe here use of an adoptive iNKT cellular therapy for acute viral infections (NCT04582201)[21].

This was an exploratory study with observational endpoints in critically unwell individuals. The initial trial of allogeneic iNKT cells (agenT-797) investigated primary endpoints of safety and tolerability. Secondary endpoints assessed evidence of improvement and resolution of ARDS following infusion, evolution of cytokine release syndrome (CRS), mortality at 30 days and 6 months and evidence of secondary infection or multiorgan dysfunction syndrome, alongside persistence of agenT-797 in vivo (Supplementary protocol files 1, 2).

Here, agenT-797 (3 patients at 100 million, 4 at 300 million, 14 at $10^9$ iNKT cells, as a single infusion) was administered to 21 mechanically ventilated patients (20 on the main trial, Table 1, plus a further critically unwell individual who received an Emergency Use Authorisation (EUA), IND number 29183; see CONSORT flow diagram, Supplementary Fig. 2) without any dose-limiting toxicities, including 5 individuals receiving VV-ECMO (Supplementary Table 1 for VV-ECMO set). Dosing was rapidly escalated as no dose-limiting treatment-emergent adverse events (TEAE) were observed, with evidence of CRS absent. Most reported TEAEs were grade 1, 2 and consistent with severe coronavirus disease-2019 (COVID-19)/ARDS (Table 2). One individual experienced a possible grade 4 TRAE of dyspnea (Table 2), a well-described sequela of SARS-CoV-2.

Signals of associated survival and prevention of secondary infections were evident, including >80% reduction numerically in pneumonia at dose cohort 3 (15%) compared to combined numbers of cohorts 1 and 2 (71%) (Supplementary Table 2). Of the 20 patients treated in the context of the main trial, 14 (70%) survived (Supplementary Fig. 3A). Survival for the 5 individuals receiving mechanical ventilation plus VV-ECMO (4 study patients, 1 EUA patient) measured 80% (4/5 alive) at 30 days and 60% (3/5 alive) at 6 months (Supplementary Fig. 3B). Here, numbers are small and thus statistical analyses and conclusions are, in the context of this small non-placebo controlled clinical trial and additional EUA patient, naturally exploratory in nature.

### agenT-797 manufacture
This is manufactured by isolation of iNKT cells from peripheral blood of healthy donors and stimulation and expansion in vitro. Multiple mechanisms have been described for both direct and indirect iNKT cell contribution to augmented anti-pathogen immunity[1–4]. agenT-797 is an HLA-unmatched allogeneic cell therapy comprised of unmodified iNKT cells formulated in chemically defined freezing medium.

**Table 1 | Patient demographics and clinical overview of main trial group by dose level cohort**

| Variable | Cohort 1 | Cohort 2 | Cohort 3 | Total | Control set[a] |
|---|---|---|---|---|---|
| agenT-797 dose level (cells) | $100 \times 10^6$ | $300 \times 10^6$ | $1000 \times 10^6$ | -- | -- |
| Patients dosed (n) | 3 | 4 | 13 | 20 | 20 |
| Age | | | | | |
| Median (range) | 67 (66–77) | 71.5 (64–75) | 62 (26–75) | 66.5 (26–77) | 70 (51–87) |
| Sex, n (%) | | | | | |
| Male | 2 (66.7) | 1 (25.0) | 7 (53.8) | 10 (50.0) | NA |
| Female | 1 (33.3) | 3 (75.0) | 6 (46.2) | 10 (50.0) | NA |
| BMI, average (range) | 32.8 (23.2–42.9) | 28.5 (16.6–40.2) | 32.6 (19.4–51.7) | 31.8 (16.6–51.7) | 30.3 (18.5–46) |
| Prior and concomitant COVID-19 medication, n (%) | | | | | |
| Received steroids (dexamethasone) | 3 (100.0) | 4 (100.0) | 11 (84.6) | 18 (90.0) | 11 (55.0) |
| Received immunomodulatory agents (tocilizumab, baricitinib) | 1 (33.3) | 2 (50.0) | 9 (69.2) | 12 (60.0) | 5 (25.0) |
| Received Remdesivir | 2 (66.7) | 4 (100.0) | 11 (84.6) | 17 (85.0) | 17 (85.0) |
| Patient disposition, n (%) | | | | | |
| Early Discontinuation | 0 | 1 (25.0) | 5 (38.5) | 6 (30.0) | -- |
| Death | 0 | 1 (25.0) | 5 (38.5) | 6 (30.0) | 19 (90.0) |

[a]Control set of intubated COVID-19 patients from single site treated between 06/2020 and 12/2020.

iNKT cells are present in healthy donors predominantly at levels between 0.01% and 0.1%, so selected healthy donors had iNKT levels above 0.05% (Fig. 1A). However, donors are selected to be homozygous for HLA-A02, maximizing the possibility of partial matching occurring with any treated patients (~40% of individuals in the US and Europe harbor an HLA-A02 allele).

The iNKT cells are isolated from peripheral blood mononuclear cells (PBMC) of healthy human donors. The manufacture is in accor-

dance with current Good Manufacturing Practice (cGMP) and comprises a continuous process with the main steps being (i) isolation of iNKT cells from PBMCs by microbead-bound monoclonal antibody to the iNKT iTCR, as mentioned[22], (ii) stimulation with the iNKT-specific ligand αGalCer-pulsed irradiated PBMCs[1–8] and (iii) interleukin-2 (IL-2) driven expansion of iNKT cells over several weeks[22]. This is followed by (iv) cell culture harvest, formulation, aseptic filling, and cryopreservation. Purity of manufactured iNKT cell products measured >99% (Fig. 1B).

iNKT cell product cytokines expressed during expansion were measured from supernatant cultures at the end of manufacturing. Both $T_h1$ (IFN-γ, GM-CSF, TNF-α) and $T_h2$ (IL-4, IL-5, IL-13) cytokines were produced (Fig. 1C). As expected for healthy donor iNKT cells[1–8], a high proportion of the iNKT cells were "polyfunctional", e.g., co-stained for $T_h1$ cytokines notably IFN-γ and $T_h2$ cytokines such as IL-4, in the same cell (Fig. 1D). Representative flow cytometric analysis of agenT-797 iNKT cell products' activation and exhaustion markers is demonstrated in Fig. 1E. The CD4+ and negative iNKT populations ('double negative' and a small variable CD8+ iNKT subset) expressed only modest variable levels of each marker. Following multiple iTCR stimulation and cytokine-driven rapid exponential expansion over a number of weeks, we observed only low-level expression of activation and exhaustion markers in the iNKT product. Only LAG-3 and GITR

### Table 2 | Adverse events

|  | Cohort 1 (n = 3) n (%) | Cohort 2 (n = 4) n (%) | Cohort 3 (n = 13) n (%) | Overall (n = 20) n (%) |
|---|---|---|---|---|
| Any AE | 3 (100.0) | 4 (100.0) | 13 (100.0) | 20 (100.0) |
| Any AE of grade ≥ 3 | 3 (100.0) | 4 (100.0) | 12 (92.3) | 19 (95.0) |
| Any TRAE | 1 (33.3) | 3 (75.0) | 1 (7.7) | 5 (25.0) |
| Any TRAE of grade ≥ 3 | 0 | 1 (25.0) | 0 | 1 (5.0) |
| Any TRAE leading to discontinuation | 0 | 0 | 0 | 0 |
| Any TRAE leading to dose interruption | 0 | 0 | 0 | 0 |
| Any TRAE leading to death | 0 | 0 | 0 | 0 |

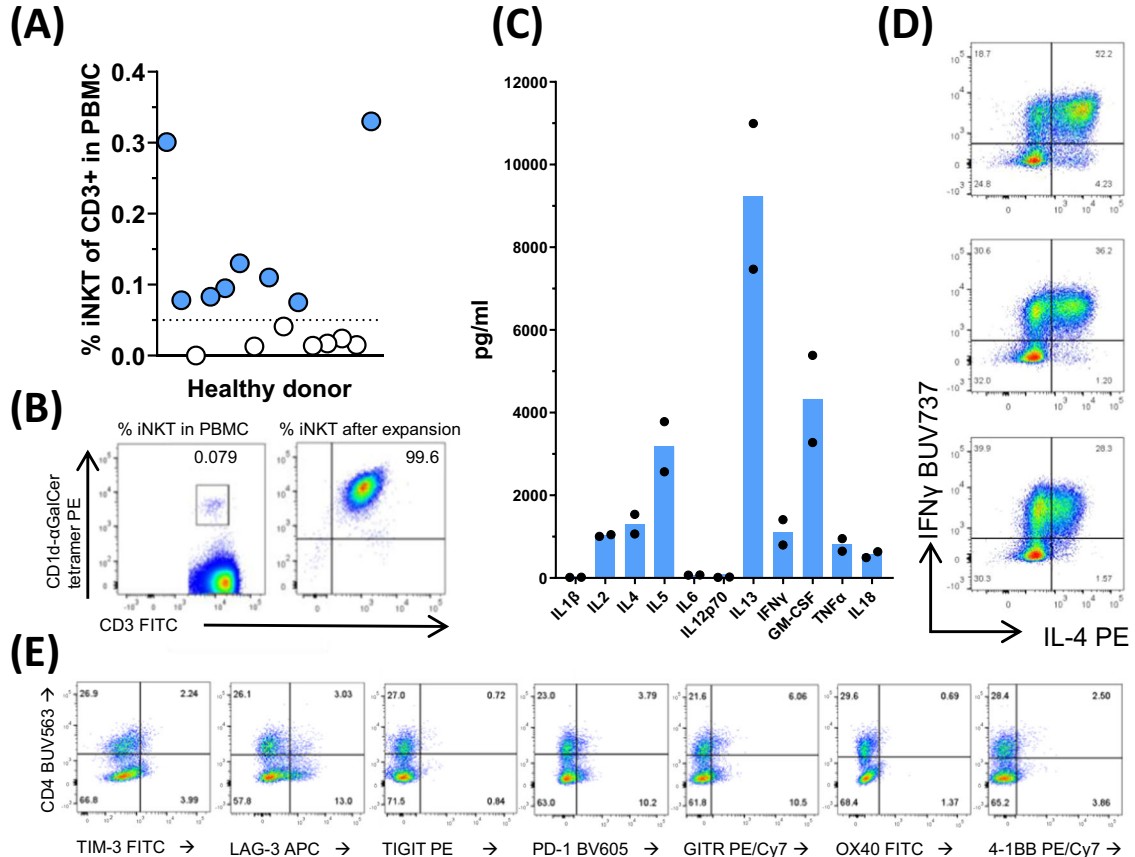

**Fig. 1 | Manufacture and characterization of agenT-797.** agenT-797 is manufactured by isolation of iNKT cells from peripheral blood mononuclear cells of healthy donors, iNKT-specific stimulation and expansion in vitro. **A** Levels of circulating iNKT cells were measured in PBMCs of 15 healthy donors and manufacturing cut-off was set at 0.05% of CD3+ lymphocytes (filled, above cut-off, open, below cut-off). **B** From selected donors, iNKT cells were purified and cells expanded using MiNK Therapeutics' manufacturing protocol. At the end of manufacturing, purity of culture measured >99%. **C** Cytokines that were produced during

expansion of iNKT cells were measured from supernatant cultures at the end of manufacturing. Both $T_h1$ (IFN-γ, GM-CSF, TNF-α) and $T_h2$ (IL-4, IL-5, IL-13) cytokines were produced. Data from two separate manufacturing runs using starting material from different donors (bars represent mean). agenT-797 iNKT cell products were subjected to flow cytometric analysis of intracellular cytokines (**D**) and surface markers (**E**) as shown. The CD4+ and negative iNKT populations expressed distinct levels of each marker. Data representative of agenT-797 batches manufactured to date (>10).

expression reached ~15%, PD-1 measurement approached 15% and others (TIM-3, TIGIT, OX40 and 4-1BB) only 1–7%.

## Patients were successfully dosed

As above, a total of 20 individuals with severe COVID-19 and receiving intensive care, were dosed in 3 cohorts between July 2020 and June 2021 (Table 1) and one additional 21-year-old patient was treated under an EUA (EUA patient) for carbapenem-resistant Pseudomonas aeruginosa pneumonia requiring VV-ECMO salvage after clearing SARS-CoV-2 infection (therapeutic schematic at Supplementary Fig. 4). The median age of the main trial cohort measured 66.5 years (range 26–77 years), the average BMI at 31.8 kg/m$^2$, the male:female sex distribution 1:1 (Table 1); all patients were intubated, on mechanical ventilation. Of the 20 treated patients, 14 (70%) survived (Supplementary Fig. 3A), compared to 10% in a comparative control (n = 20) evaluated at the same institution. Though increased BMI is a risk factor for adverse outcomes after infection with SARS-CoV-2, no significant difference in BMI between patients who survived (average 32.1; range 16.6–51.7) and those who died due to COVID-19 (average 31.2; range 19.4–40.2) was observed.

As treatments were being developed in such sick individuals with this 'new disease' in real time, it is difficult to ascribe adverse events to treatment(s) or the disease itself. In our clinical experience, agenT-797 appeared well tolerated, with the majority of adverse events (AEs) consistent with underlying disease. Most reported TEAEs were grade 1, 2 and consistent with severe COVID-19/ARDS including anemia (n = 8), fever (n = 7), and acute kidney injury (n = 6). One patient experienced a possible grade 4 TRAE of dyspnea (Table 2), again difficult to ascribe this to treatment during a pandemic.

There were 2 urinary tract infections noted on the day of infusion which resolved within 3 and 16 days respectively. There was one pulmonary infection (Methicillin Resistant Staphyloccocus Aureus, MRSA) noted on the day of infusion, which resolved within 5 days of infusion. All other infections secondary infections occurred post infusion of agenT-797. There was a potential dose-dependent reduction in the occurrence of secondary infections observed, including an 80% reduction in pneumonia at dose cohort 3 (15%) compared to combined numbers of cohorts 1 and 2 (71%) (Supplementary Table 2). Although highly intriguing and consistent with reported anti-infectious properties of iNKT cells[1–4], numbers are small with respect to these conclusions. We rapidly escalated to the highest dose cohort, as we considered there to be a lack of toxicity related to agenT-797. Indeed, the EUA patient was treated at the highest dose level and their carbapenem-resistant Pseudomonas pneumonia rapidly cleared post agenT-797 infusion, supporting our view that agenT-797 might be able to assist in the control of secondary infections.

Of the 21 agenT-797 treated patients (including the EUA patient), 5 were given agenT-797 during VV-ECMO, traditionally regarded as 'the most aggressive salvage therapy' for critically ill individuals with COVID-19 (with one systematic review showing a 41% mortality in VV-ECMO treated patients[23]). Survival of the VV-ECMO cohort was 80% (4/5) at 30 and 90 days and 60% (3/5) at 120 days. This compares favorably to overall survival of 51% (18/35) for patients with COVID-19 treated with VV-ECMO in the same institution during the same timeframe (January 2021 to January 2022, Supplementary Fig. 3B, Supplementary Table 1). In patients treated with agenT-797, we did not observe cell therapy-associated oxygenator failure due to clotting in filters, as reported routinely with mesenchymal stem cell therapy in ARDS patients on VV-ECMO[24]. Here, agenT-797 represents the first use of immune cell therapy in patients receiving VV-ECMO.

## agenT-797 induces an anti-inflammatory signature without CRS

Acute phase clinical parameters and cytokines were measured in sera of patients receiving agenT-797 over a 28-day period (Fig. 2). We measured key indicators of CRS, notably IL-1α, IL-1β, IL-6, ferritin,

C-reactive protein (CRP) and D-dimer levels. There were no significant rises in these (Fig. 2A). IL-6, ferritin, CRP and D-dimer elevated in patients before infusions, but levels decreased post-infusion (Fig. 2A). It is difficult to delineate whether this is due to the natural history of the disease or agenT-797.

We also measured an extensive panel of serum biomarkers. Of these, IL-1RA (Fig. 2B) demonstrated the most significant changes in serum levels post-infusion of agenT-797, consistent with an increased anti-inflammatory response counteracting IL-1 mediated cytokine release. Conversely, levels of the pro-inflammatory cytokine IL-7 reduced significantly.

For the EUA patient, we evaluated changes in cytokine profile in the lungs through bronchoalveolar lavage (BAL) alongside sera (Fig. 3). Infusion of agenT-797 was followed by a reduction within 24 h within BAL of key pro-inflammatory cytokines TNF-α and IL-1β as well as of the myeloid cell-suppressing cytokine IL-10, all of which were sustained. This rapid change in the BAL cytokine profile was followed by secondary changes over the next five days, including a transient increase of the T cell-stimulating cytokine IL-12 (p70), a transient decrease in the IFNγ-inducing cytokine IL-18 and a sustained increase in the eosinophil recruiting and activating factor, IL-5. Further delayed changes were observed by hospital day 36 (HD36; infusion day 12, ID + 12), including an increase in anti-inflammatory cytokines (IL-1RA, IL-4) as well as activators and chemo-attractants of lymphocytes and neutrophils (IL-6, IL-7, IP-10, IL-8). Most post-infusion changes in the cytokine profile remained local to the lung (BAL).

## Persistence of agenT-797

We measured persistence of agenT-797 in the patients' PBMCs, by digital droplet PCR based on genetic markers unique to donor material. As shown (Fig. 4), agenT-797 is detected in the peripheral blood up to day 6 post-infusion, with notably higher levels and likely low-level longer persistence detectable at the highest dose level, up to the last day of sampling at day 28. Peak levels of agenT-797 demonstrate a dose-proportional relationship. The red line indicates patients treated with agenT-797 and on VV-ECMO, and the detection of agenT-797 for prolonged periods post-infusion in VV-ECMO patients is consistent with the aforementioned lack of filter clogging in VV-ECMO patients treated here.

White blood cell (WBC) parameters (Supplementary Fig. 5), reflect COVID-19 ITU patients with a variety of secondary infections (Tables S2), as found in general, particularly at this time of the pandemic[25]. Post-infusion, there did not appear to be major changes in the composition of WBCs beyond presumptive infection-stimulated spikes and reciprocal troughs. However, eosinophils did show a delayed potentially transient increase in WBC in all cohorts ~8 days post-infusion, as shown (Fig. S5B, C). A prior study showed that the percentage of endogenous iNKT changes reflect absolute numbers in COVID-19 patients[22]. In the current trial, substantial numbers of new healthy donor iNKT cells were infused, up to approximately the equivalent of an individual's own existing iNKT repertoire at the highest dose.

As iNKT cells are tissue-resident lymphocytes, we measured the dynamics of tissue distribution of agenT-797 in a murine xenograft model (Supplementary Fig. 6), demonstrating rapid translocation of agenT-797 to tissue following intravenous injection, as previously[22]. Though the tissue distribution of agenT-797 in patients remains difficult to assess, the observed transient post-infusion persistence of agenT-797 in patients' blood samples remains consistent with these preclinical dynamics of blood-to-tissue distribution of agenT-797 in vivo.

Infusion of agenT-797 into the EUA patient appeared to induce a shift in the cellular composition within BAL (Fig. 4B). A sustained decrease in the fraction of neutrophils within BAL was observed post-infusion of agenT-797, accompanied by a transient increase in the

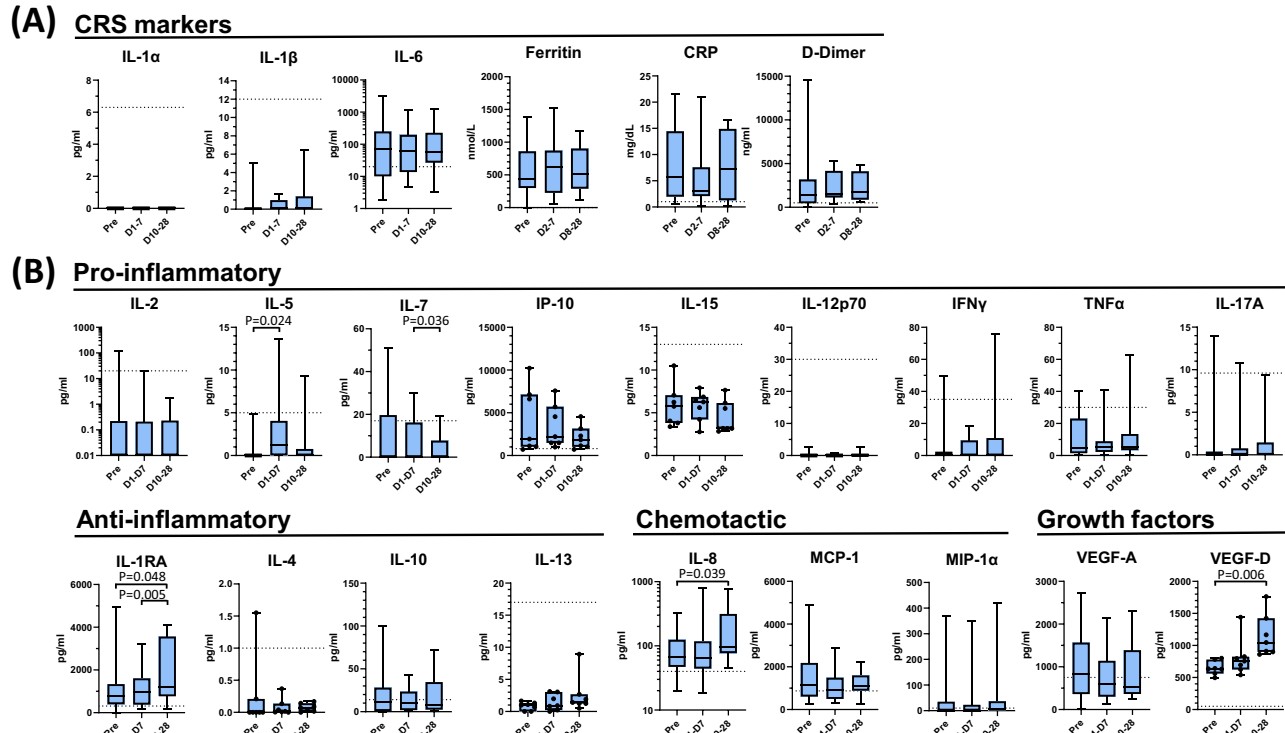

**Fig. 2 | Post-infusion changes in serum biomarkers.** Serum cytokine levels broken into (**A**) CRS markers and (**B**) functional classes from pre- and post-infusion timepoints indicated. Pre-infusion serum sample was taken on day of infusion with agenT-797 (ID0 Pre). Cytokines were measured by Luminex using a custom multiplex assay (ThermoFisher). **A** Serum levels of key indicators of CRS; IL-1α, IL-1β, IL-6, ferritin, C-reactive protein (CRP) and D-dimer levels. Biomarker concentration determined over 28 days. **B** Serum cytokine levels of selected biomarkers spanning the immuno-regulatory spectrum. Cytokine data is binned as described below. Dotted lines indicate upper limit normal level of respective biomarker in healthy people. Cytokine data are grouped into pre-infusion sample (pre, single timepoint on day 1), early post-infusion window (average from 2 h post infusion on day 1 to day 7 post-infusion; D1-7) corresponding to the measured persistence of agenT-797 in the periphery, and a late post-infusion time window (days 10–28; D10-28). Data

for ferritin, CRP, D-dimer shown for pre-infusion timepoint on day of treatment (pre, singe timepoint), early post infusion window (days 2–8; D2-8) and late post-infusion time window (days 8–28; D8-28). IL-1α/β not detected or within normal range. Cytokine data from all patients (*n* = 20), except for IL-4, IL-13, IL-15, IP-10 and VEGF-D where data are available only for cohorts 1 and 2 (*n* = 7). Data for ferritin, CRP, D-dimer measurements from 11 patients (Cohort 1: *n* = 3; Cohort 2: *n* = 2; Cohort 3: *n* = 6). All sample measurements returned as under the limit of detection scored as zero. All significant changes (adjusted *p* value < 0.05) post dosing are indicated (column means comparison using ANOVA/mixed effect model followed by Tukey's multiple comparisons test). Box plot whiskers extends to the minimum and maximum values, with the box encompassing the interquartile range (IQR), and the center line indicating the median.

fraction of eosinophils, followed by an increase in the percentage of macrophages/monocytes. Lymphocytes compromised a minority fraction of nucleated cells within the BAL and their frequency remained relatively unchanged post-infusion of agenT-797. agenT-797 could be detected in EUA patient peripheral blood, peaking at day 1 post-infusion (ID + 1), but not in the BAL post-infusion, within the sensitivity of the droplet PCR assay used to distinguish the exogenous iNKT from the patient's own iNKT cells (Fig. 4C).

### Treatment induces transient donor specific allo-antibodies (DSA)
agenT-797 induced measurable DSA by day 14 post-infusion. The incidence of DSA development was reduced with increased HLA class I-matching but appeared unrelated to the degree of HLA-class II matching (Fig. 5A). In Fig. 5B, serum levels of DSA post-dosing were reduced with increased degree of HLA matching. DSA levels post infusion of agenT-797 peaked at day 14 and appeared to decrease in 3/4 patients following this (Fig. 5C).

### agenT-797 activates both adaptive and innate immune systems
To delineate how iNKT cells can potentially modulate local and systemic immune responses we developed co-culture assays to investigate the interaction with myeloid lineage cells, which express CD1d, the natural ligand for the invariant iNKT cell iTCR. Specifically,

dendritic cells and macrophages are key players in the innate response to infections. As shown (Fig. 6A), agenT-797 activates dendritic cells, which in turn leads to further iNKT cell activation (Fig. 6B). agenT-797 also directly interacts with both "M2" and "M1"-type macrophages, as evidenced by an increase in iNKT cell activation during co-culture with both macrophage subtypes (Fig. 6C, D). We observed that M2 macrophages lead to a higher level of iNKT cell activation and cytotoxic response than M1 macrophages (Fig. 6D, E). Combined, these features indicate a high level of cross-talk between iNKT cells and myeloid lineage cells, which may be a key contributor to the observed clinical responses, specifically the reduction of secondary infections.

Whereas conventional NY-ESO TCR-positive T cells become exhausted after repeated cycles of antigen exposure (Fig. 7A–D), agenT-797 improves the cytotoxic capacity, activation and cytokine production of partially exhausted T cells (Fig. 7E–I) via soluble factors (Fig. 8A–C).

## Discussion
iNKT cells have been under investigation as immunotherapeutic agents for two decades[1–7], mainly in cancer, due to their potent anti-tumor cytotoxic ability. Immunotherapies utilizing iNKT cells have potential to similarly treat critically unwell patients with acute respiratory infections, as in the current setting we describe at the height of the COVID-19 pandemic. We show the pre-clinical activity of

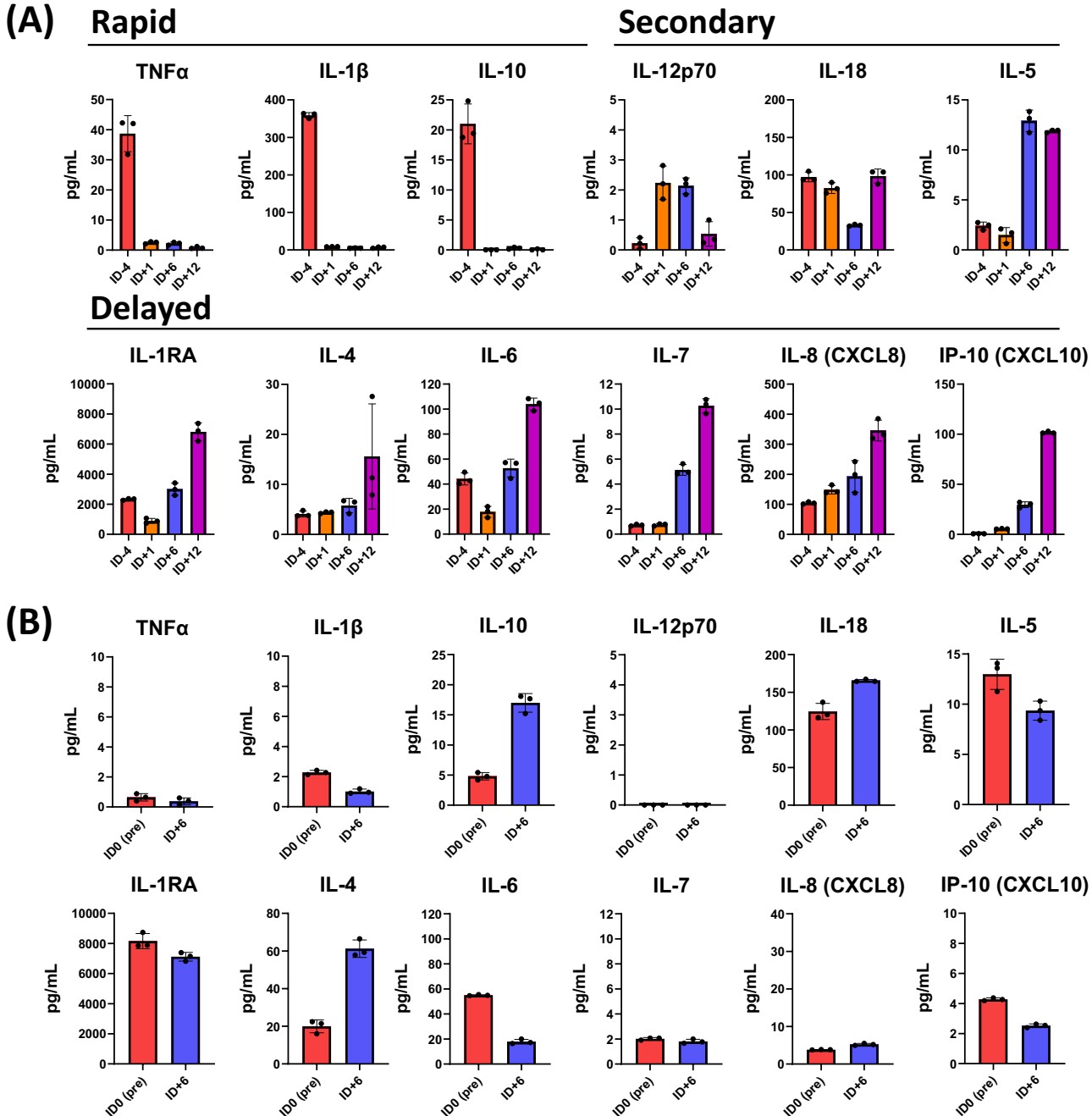

**Fig. 3 | Post-infusion cytokine level changes in BAL and serum.** Cytokine levels measured in EUA patient broken into onset classes (Rapid: onset and durable change within first day post-infusion; Secondary: onset and peak response within first week of infusion; Delayed: onset and peak following first week of infusion) in (**A**) BAL versus (**B**) sera at pre- and post-infusion timepoints indicated. For sera, the pre-infusion serum sample was taken on day of infusion with agenT-797 (ID0 Pre). Cytokines were measured by Luminex using a custom multiplex assay (Thermo-Fisher). Bar colors: Pre-infusion (red); Day post infusion (orange); Day 6 post-infusion (blue): Day 12 post-infusion (purple). Only pre- and a single post-infusion point (ID + 6) were available for sera. Measurements in triplicate, bar represents mean with SD.

allogeneic iNKT cells (agenT-797) and safety as an adoptive cellular transfer drug during the SARS-CoV-2 pandemic, for critically unwell patients with ARDS. This has potential broad applicability in various severe infections, and it is notable that we were able to rapidly scale allogeneic iNKT cell GMP-standard manufacturing during the pandemic and successfully dose patients in a variety of clinical settings, including a community hospital without a dedicated cell processing facility.

Mechanistically, we demonstrated that agenT-797 can rescue exhausted T cells via soluble factors (Fig. 8) and activate dendritic cells

as well as preferentially killing tumor- and infection-promoting M2 macrophages over M1 macrophages, creating in theory a hostile environment for both tumor and infected cells (Fig. 6). 'M2' cell lysis might be expected to bias towards more pro-inflammatory 'T_h1' type responses, which can be anti-tumor and anti-pathogen. However, M1 and M2 are not 'black & white' definitions. Indeed, the strongest serological finding of substantial IL-1RA increases purports to show a different picture (Fig. 2B). Moreover, lack of potent impact on CRS cytokines and factors (Fig. 2A), suggests iNKT cells act beneficially downstream of these, presumably via other (cellular) players.

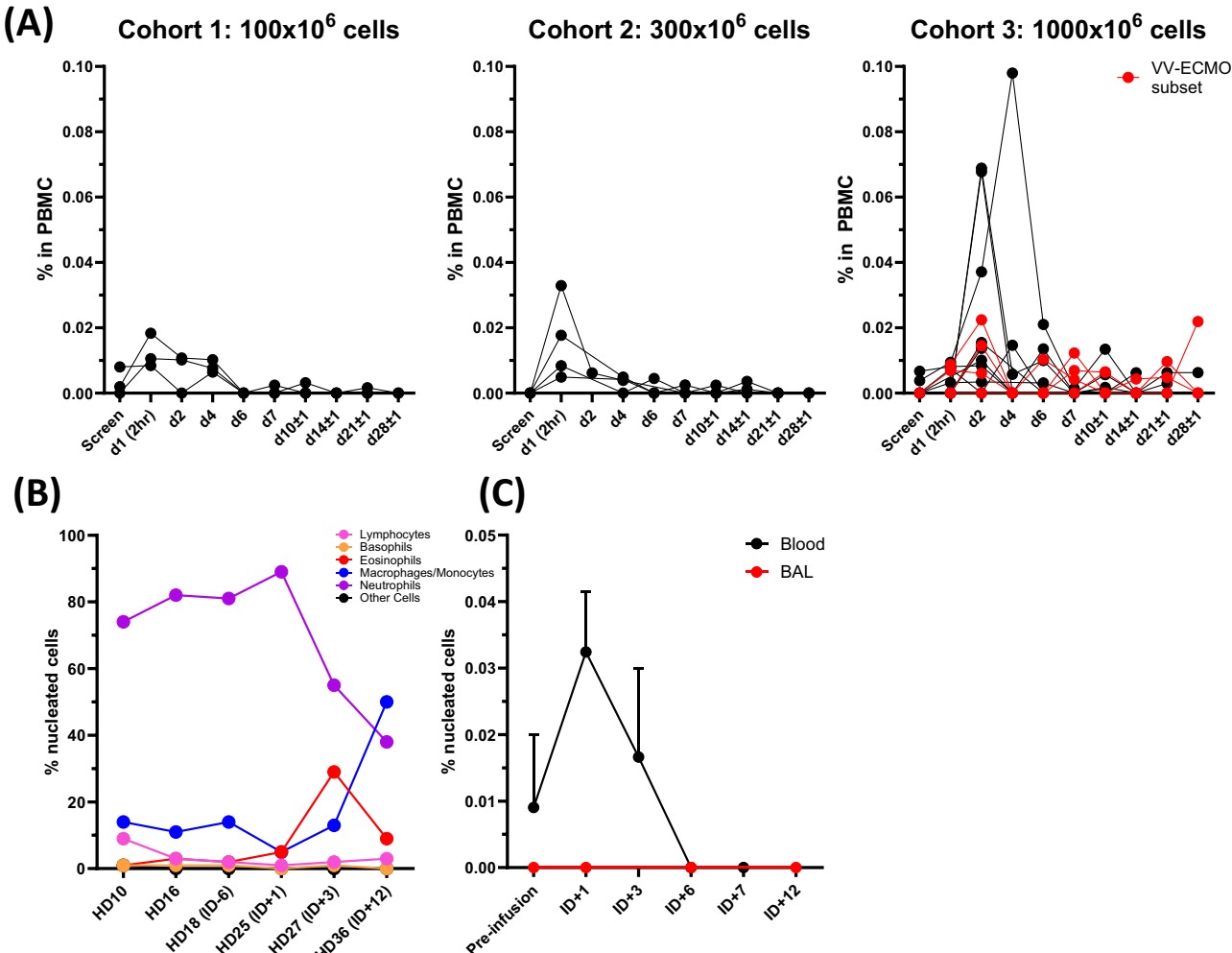

**Fig. 4 | Detection of agenT-797 at multiple timepoints during patient hospitalization.** Cellular composition of nucleated material in blood and BAL. Whole blood frozen at −80 °C and sent to MiNK for evaluation. BAL samples collected at bedside and immediately separated into two aliquots, then one aliquot was immediately frozen at −80 °C and sent to MiNK for evaluation. The remainder of the sample was sent to the clinical lab for processing. Cell counts and differentials were determined and closely correlated. **A** Persistence of agenT-797 in blood. Cohort-level peripheral persistence of agenT-797 Quantification of agenT-797 in patient PBMC by digital PCR based on genetic markers unique to donor material. Each line represents data from one patient. The red line indicates patients treated with agenT-797 and on VV-ECMO. **B** EUA patient. Standard cytology cell counts in BAL and differentials. **C** EUA patient. Detection of agenT-797 in peripheral blood (black circles) and BAL (red circles) pre-infusion and at various timepoints post infusion with agenT-797. Data for four separate agenT-797-specific markers measured at each timepoint in blood and BAL samples shown as mean with standard deviation (SD). Detection and quantification of agenT-797 was by detection of agenT-797-specific single nucleotide polymorphisms (SNPs) using a digital droplet (dd) PCR-based assay to detect genetic chimerism (Neogenomics, CA). Pre-infusion timepoint for BAL: ID -4, and for blood: ID 0.

Interestingly, eosinophils showed a delayed rise in WBC in most patients of all cohorts after ~8 days post infusion, and we note that this rise was preceded by a transient increase in serum of the eosinophil-recruiting and activating cytokine IL-5 (Fig. 2B). These observations extend to the lung, where in the EUA patient. we note a transient increase of eosinophils in BAL concordant with an increase in BAL IL-5 (Figs. 3A, 4B). Extensive further work in complex models with knockouts of cytokine and other genes would be required to fully elucidate precise mechanisms involved.

Here, agenT-797 represents a variant-agnostic approach to COVID-19-induced ARDS and is the first iNKT adoptive immunotherapy to be described in the clinic, and to our knowledge is the first immune cell therapy of any type to be administered safely to critically unwell patients receiving VV-ECMO. Despite high ordinal scales (OS 7) indicative of a poor prognosis, patients treated with agenT-797 at 3 U.S. medical centers in this study demonstrated favorable outcomes, with an on-study 30-day survival of 70% versus 10% in a control set and 39% in the Centers for Disease Control and Prevention (CDC) hospital

survival outcome set, for the same time period. Additionally, we included BAL and blood cytokine data in the context of an active *post*-Covid-19 Pseudomonas infection in the agenT-797 treated EUA patient, showing in principle potential generality to both concomitant and sequential viral & bacterial infections. While consistently beneficial in this small cohort, we are reluctant to draw definitive efficacy conclusions regarding agenT-797 here, given the small numbers of patients and lack of randomization.

We observed an increase post-infusion in IL-1RA, an anti-inflammatory marker. In contrast, there was a decrease in IL-7 and other pro-inflammatory cytokines and other clinical markers (Figs. 2 and 3), data consistent with agenT-797's ability to stimulate certain myeloid cells (Fig. 6) and to reinvigorate partially exhausted T cells described above (Figs. 7, 8)[26]. We also observed transient persistence in the periphery, with peak levels correlating with dose level as well as an anti-drug antibody response that appears short-lived, together (Figs. 4 and 5) suggesting potential for redosing. A reduced expected incidence of pneumonia was observed in patients treated at the

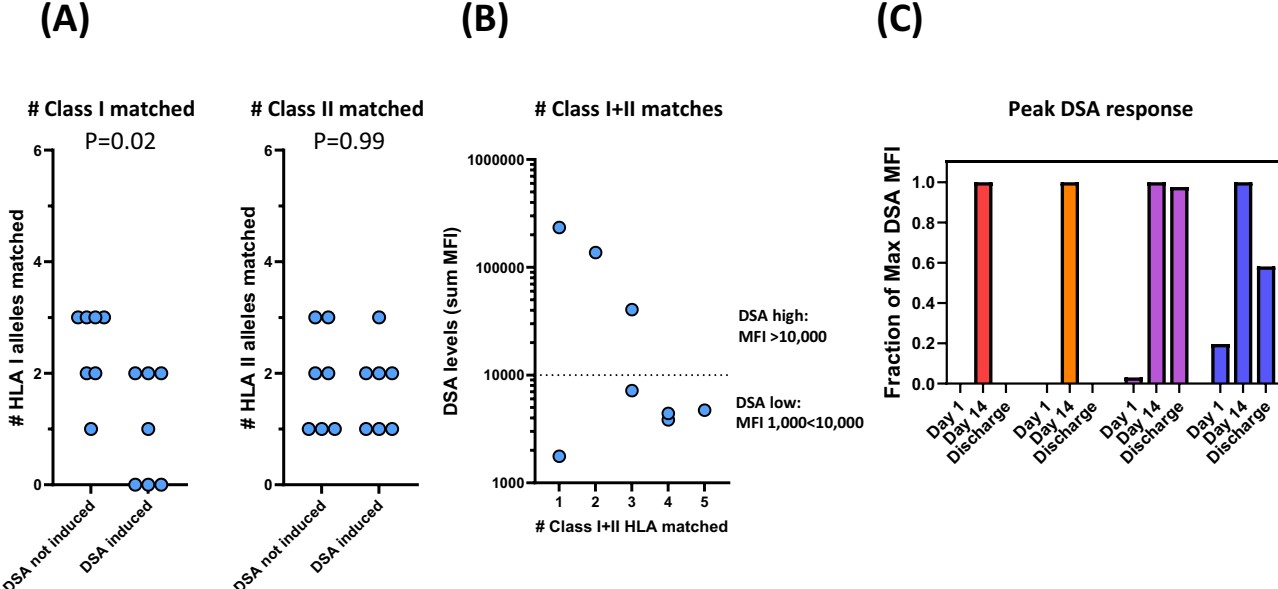

**Fig. 5 | Post-infusion induction of donor specific allo-antibodies (DSA).**
**A** Presence of donor specific allo-antibodies (DSA) was determined on day of dosing and day 14, showing induction of DSA. Patients were scored weather DSA were induced or not induced at day 14 post infusion. Small changes (<1000 MFI) in DSA measurements between D1 and D14 were scored as not induced. Incidence of DSA development is reduced with increased HLA class I matching but appears unrelated to the degree of HLA-class II matching. Mean column values compared using Mann–Whitney nonparametric test, with $P$ values (two-tailed) indicated. **B** Serum levels of DSA post-dosing are reduced with increased degree of HLA matching. DSA levels of combined MFI < 1000 are considered negative and not reported. **C** For 4 patients (identified by different colors) DSA levels were measured at the time of discharge (day of discharge of patients, from left to right: 60, 28, 21, 28). DSA levels post infusion of agenT-797 peak at day 14 and appear to decrease afterwards (data normalized to peak DSA levels for each patient).

highest dose, again underscoring the potential application of agenT-797 in viral diseases and infections more broadly (Supplementary Table 2). In the EUA patient, infusion of agenT-797 was accompanied by significant, localized (BAL) changes in the cytokine profile (Fig. 3), likely related to a complex remodeling of the immune environment within the lung (Fig. 4), positively affecting immune cell activation and trafficking. It is possible that iNKT cells can play a regulatory role and directly decrease pulmonary inflammation, potentially contributing to the highly favorable safety profile we observed.

As well as a potential lack of toxicity, a number of features of iNKT cells make them potentially useful as adoptive immunotherapy[1–7] (Supplementary Fig. 1). iNKT cells recognize both foreign lipid antigens and self-lipid antigens. The iTCR-lipid-CD1d interaction is similar for both self and foreign lipid antigens, despite the differences that exist in these lipid structures, and production of lipid self-antigens for iNKT cells can be upregulated by antigen-presenting cells (APC) in response to danger signals, such as Toll-like receptor (TLR) agonists. This in turn provides a mechanism for iNKT cell activation during an acute infection, even in the absence of foreign lipid antigens. In addition to being activated through their iTCRs in response to CD1d-presented lipids, iNKT cells can also be activated by other stimuli, such as pro-inflammatory cytokines, again of relevance here as viral levels may have decreased by the time of the infusion. During many infections, interleukin-12 (IL-12) may have an equally important role to lipid antigens in activating iNKT cells. In aggregate, iNKT cells couple the rapid activation kinetics of innate immune cells with the diverse effector functions of adaptive T cells and as shown here, early activation during infection leads to rapid cytokine production in target tissues by iNKT cells. Interactions between iNKT cells and CD1d-expressing APCs lead to bidirectional activation: cytokines produced by iNKT cells are thought to activate and recruit other cell types early during immune responses, while activated APCs direct the ensuing adaptive immune responses. Thus, iNKT cells and their lipid antigens help to orchestrate both innate and adaptive immune responses[1–7].

Specifically, we now propose that the iNKT interaction with M1 macrophages induces pro-inflammatory cytokine secretion (from both cells), and interaction with M2 macrophages resulting in an anti-inflammatory cytokine signature. In a physiologic setting, the inter-action is likely more nuanced, and one can speculate that agenT-797 in the inflamed lung modulates the activity of both M1 and M2 macrophages, resulting in complex changes in lung cytokines, as observed (Fig. 3). Assuming the changes we observe in lung cytokines derive largely from the iNKT-macrophage interaction, one could argue that agenT-797 acts to rapidly reduce secretion of cytokines from both M1 (TNFα, IL-1b) and M2 (IL-10) macrophages, followed by a modified cytokine signature derived from M1 macrophages (IL-6, IL-8, IL-12, IP-10), M2 macrophages (IL-4), CD169+ macrophages (IL-1RA), and iNKT cells (IL-4, IL-5), consistent with previous cellular interaction data[27–29].

NK cells use receptors such as CD16 and NKG2D to recognise and subsequently kill virus-infected cells by releasing perforins and gran-zymes, which induce target cell apoptosis and lysis. They also produce IFN-γ which aids in mobilizing APCs as well as promoting the devel-opment of effective antiviral immunity[25,30]. However, during SARS-CoV-2 infection, a reduction in NK cells has been observed with decreased cytolytic function observed in remaining NK cells[31–34]. Pre-vious data have shown that patients outside of the intensive care unit (ICU) had significantly higher levels of NK cells compared to those in ICU[35,36], supported by another study suggesting an impairment of immune cytotoxic effector functions[35,36]. iNKT cells can trans-activate NK as well as conventional T cells[1–7], so this represents another anti-viral mechanism of relevance in the current context.

In moderate cases of COVID-19, previous research has shown that an expansion of CD160+ 'NKT-type' cells has been observed, which may help promote rapid resolution of infection through direct cytotoxicity, but the NKT CD160 cluster was notably absent in severe COVID-19 cases[37]. The authors of the aforementioned work[37] did not specify whether the NKT cell cluster were type I (αGalCer-reactive iNKT cells)

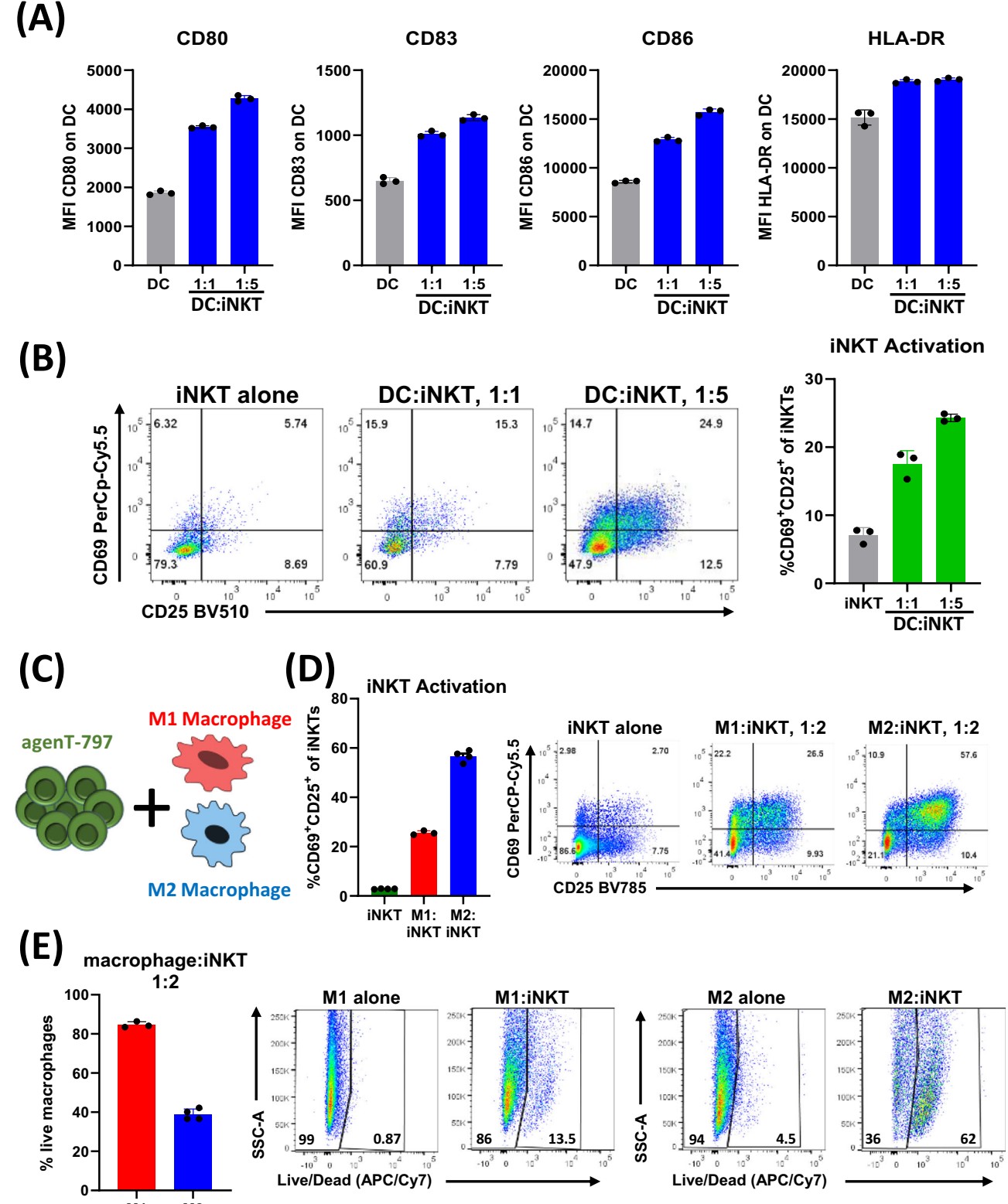

**Fig. 6 | Cross talk between agenT-797 and the myeloid cell compartment.**
**A**, **B** DCs were cocultured with agenT-797 cells at 1:1 and 1:5 DC to iNKT ratio for 48 h. DC and iNKT activation was measured by flow cytometry. Data shown is representative from one donor. **A** Expression of DC activation markers CD80, CD83, CD86 and HLA-DR on DCs cultured alone (gray bars) or with iNKT cells (blue bars; *n* = 3). **B** Evaluation of iNKT activation from DC co-cultures by co-expression of CD25 and CD69 (*n* = 3; iNKT alone: gray bars; coculture: green bars). **C**–**E** M1 and M2 macrophages were cocultured with agenT-797 at 1:2 ratio for 48 hr as shown in

schematic (**C**). Data shown is representative from one donor. **D** iNKT (agenT-797) activation following macrophage coculture as measured by surface expression of CD69 and CD25. agenT-797 alone (green; *n* = 4) or cocultured with M1 (red; *n* = 3) or M2 (blue; *n* = 4) macrophages. Representative flow chart shown on left.
**E** Frequency of viable macrophage following coculture of iNKT cells with M1 (*n* = 3) or M2 (*n* = 4) macrophages as measured by live/dead staining. Representative flow chart shown on left. All bar charts represent mean with SD.

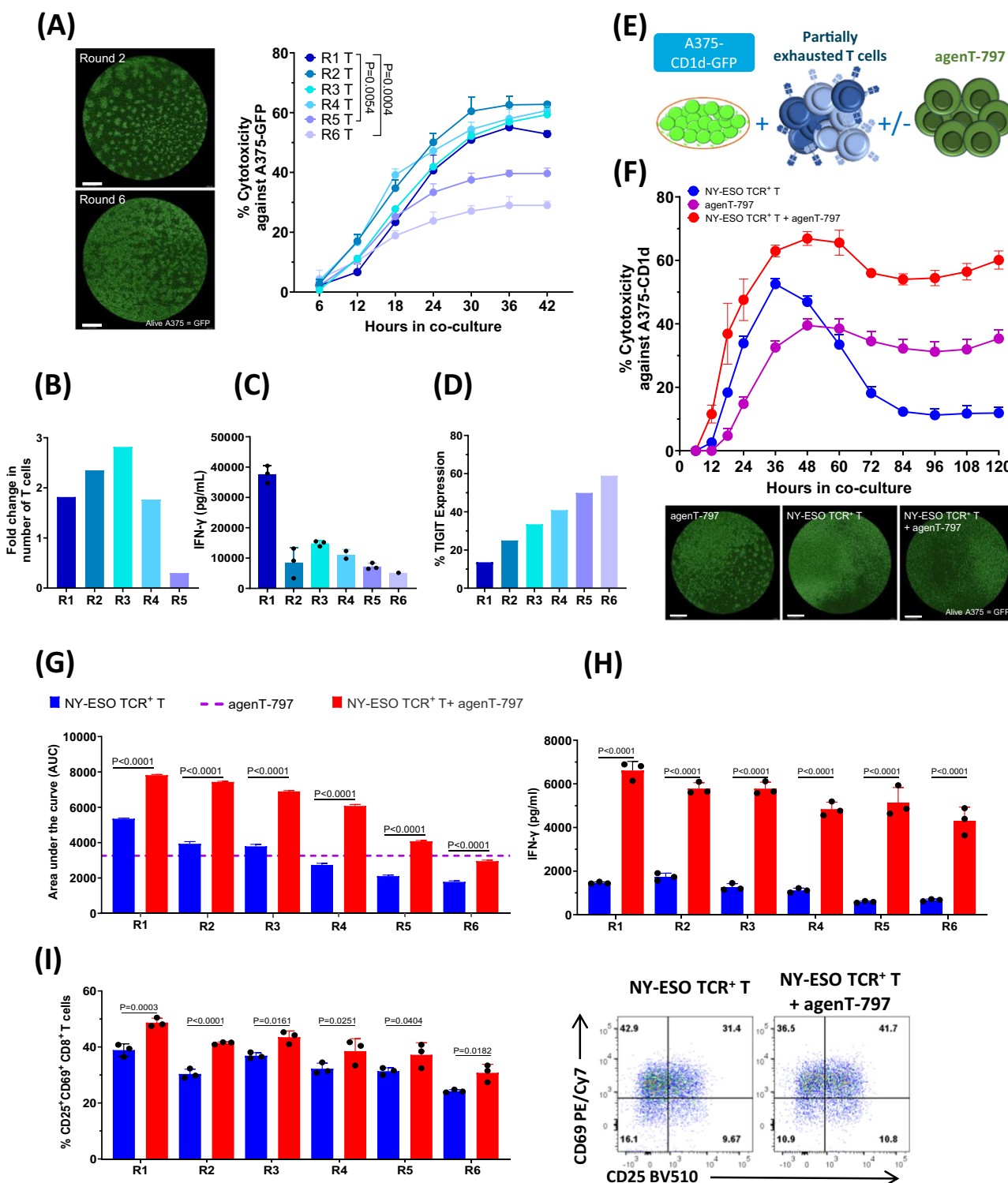

or type II NKT cells[1–3] and CD160 is expressed by both subtypes[38]. Our manufacturing process utilizes αGalCer-induced stimulation producing a purely iNKT cell product, likely without contribution from type II NKT cells. In severe cases of COVID-19, circulating iNKT cells have been shown to be activated by IL-18, which is a cytokine associated with unconventional T cell activation during viral infections in general. Furthermore, there is a decrease in circulating iNKT cells and in iNKT cell production of IFN-γ in COVID-19[39]. Of the remaining iNKT cells in circulation, levels of those expressing PD-1 and CD69 were increased, while high PD-1 expression persisted on iNKT cells from patients in ICUs at day 15[39]. Overall, transiently restoring iNKT cells from a healthy donor appears a safe and promising strategy in an acute setting.

Another team has reported the development of allogeneic chimeric antigen receptor (CAR)-iNKT cells as a potential adoptive immunotherapy in B cell malignancies, with an ongoing trial (NCT05487651). The data presented here of allogeneic unmodified iNKT cells in a viral infection are unrandomized and numbers are small, precluding formal efficacy analysis. However, we have shown that this approach can be rapidly scaled and apparently safely used in critically unwell individuals; randomized trials of this approach are warranted.

**Fig. 7 | Reversal of partial T cell exhaustion by agenT-797. A** Cytotoxicity of NY-ESO TCR + T cells against A375-GFP cells was assessed using Incucyte Live Imaging (representative images after 48 h of co-culture; scalebar: 1 mm). Significant differences in killing curves between single-time antigen-exposed (Round 1; R1) and multiple-times antigen-exposed (R5, R6) T cells became apparent after 30 h of coculture. Adjusted *P* values shown for data from 30 h timepoint. Data analysed by 2-way ANOVA followed by Dunnett's multiple comparison test. Datapoints represent mean with SD (*n* = 3 replicate wells). **B** T-cell proliferation (fold change) during each round of antigen exposure (*n* = 1 experiment). **C** IFN-γ measurement in supernatants after each round of antigen exposure of T cells. Bars represent mean with SD (*n* = 3). **D** Expression of TIGIT in CD8 + T cells evaluated as a marker of exhaustion in every round of exposure. Data shown is representative from one donor. **E–I** NY-ESO TCR + T cells from different rounds of antigen exposure were co-cultured with A375-CD1d-GFP target cells in presence or absence agenT-797 cells

at the ratio of 1:1:1 as shown in the schematic in (**E**). **F** Impact of agenT-797 on killing curve of Round 4 (R4) antigen-exposed T cells. Datapoints represent mean with SD (*n* = 3). Sample images from end of experiment (scalebar: 1 mm). **G** Impact of agenT-797 on cytotoxic capacity (area under the curve; AUC) of T cells with 1–6 rounds of prior antigen-exposure (R1-R6). Bars represent mean with SEM (*n* = 3). **H** Impact of agent-797 in IFN-γ release in supernatants from antigen-exposed T cells (R1-R6). Bars represent mean with SD (*n* = 3). **I** Impact of agenT-797 on activation of antigen-exposed CD8 T cells (R1-R6) as measured by CD25 and CD69 expression. Bars represent mean with SD (*n* = 3). Flow panels: Impact of agent-797 on expression of activation markers (CD69 + CD25 + ) on Round 4 (R4) antigen-exposed T cells. **H, I** For each set of antigen-exposed T cells (R1-R6), mean values of respective T cell response (cytotoxicity AUC, IFN-γ release, CD25/69 expression) in absence or presence of agenT-797 were compared by Šídák's multiple comparison test. *P* values adjusted for multiple comparisons. Colors as in (**G**).

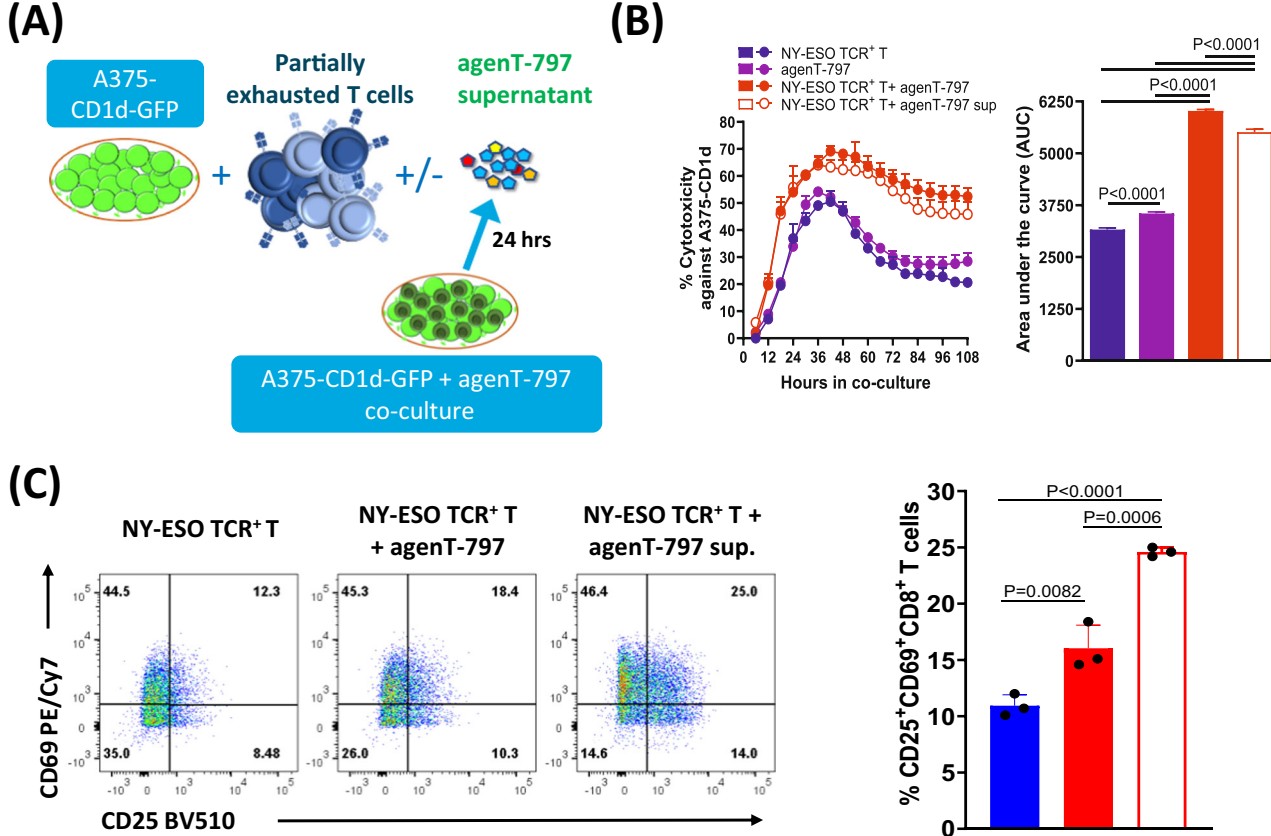

**Fig. 8 | agenT-797 acts via soluble factors to rescue partially exhausted T cells.** Supernatant from agenT-797 co-culture with A375-CD1d-GFP cells was collected after 24 h and added to coculture assays for antigen-exposed NY-ESO TCR + T cells as shown in the schematic in (**A**). **B** Impact of agenT-797 supernatant (sup) on cytotoxicity of Round 2 antigen exposed T cells against target cells. Left panel: killing curves over 5 days. Datapoints represent mean with SD (*n* = 3). Right panel: Cytotoxicity as measured by area under the curve (AUC). Bars represent mean with

SEM (*n* = 3). **C** Impact of agenT-797 supernatant an activation of Round 2 antigen-exposed T cells measured by CD25 + CD69+ expression after 24 h of co-culture. Bars in bar graph represent mean with SD (*n* = 3). Colors represent conditions as in (**B**). **B, C** Supernatant from activated agenT-797 significantly improved the cytotoxicity and activation of partially exhausted T cells. Column means compared using Šídák's multiple comparisons test. *P* values adjusted for multiple comparisons.

Emerging studies into understudied unconventional T cell subsets in COVID-19 reveal intriguing information implicating different subsets in involvement in both pathogenesis and resolution of infection, though the underlying mechanisms and stage-dependent functions remain largely undefined. Thus far, data suggests that unconventional T cell subsets such as iNKT cells become activated in response to viral infections such as SARS-CoV-2 and there is evidence for protective antiviral responses. The data here suggests that such effector functions can be successfully harnessed for therapy.

iNKT cells engineered to express various CARs in preclinical studies can enhance tumor killing as well as target tumor stroma, reduce

tumor microenvironment immunosuppression and promote infiltration of tumor specific CD8 + T cells[40,41]. Our allogeneic native iNKT cell therapy, agenT-797, continues to advance in further Phase 1/2 clinical trials alone and in combination (with anti-PD-1 immunotherapy) in solid tumors (NCT05108623) and multiple myeloma (NCT04754100). It seems likely that such unconventional T cells become activated in response to cellular or inflammatory stimuli rather than through direct viral or tumor recognition (Supplementary Fig. 1). This raises the potential for future targeted therapeutic manipulation of these cells using exogenous cancer-, microbe-derived-, metabolite- or cytokine-based stimuli. In addition to

enhancing understanding of unconventional T cells in primary SARS-CoV-2 infection, loss or dysfunction of such cells may also increase susceptibility to other microbial infections, driving secondary infections which can lead to serious complications such as ARDS/sepsis.

There is a need to learn more, particularly in the clinical setting, about the complex antimicrobial effector functions of unconventional T cell subsets such as iNKT cells, cells which can bridge protective innate and adaptive immunity. A recent related preclinical COVID-19 study and clinical results of partly iNKT-acting adenosine A2A receptor agonist (Regadenoson) in COVID-19[42,43], as well the current clinical study provide compelling evidence to support randomized clinical studies of iNKT cell-based therapy in severe acute infectious settings including critically unwell individuals, as described here.

## Methods

### Patients

This was a phase 1/2 study to evaluate the safety and potential efficacy of agenT-797, an unmodified, allogeneic iNKT cell therapy, in participants with moderate to severe ARDS secondary to SARS-CoV-2 or influenza, either with intubation or at high risk to be intubated, as determined using Berlin definition[44] (trial registration NCT04582201). This was a standard 3 + 3 dose escalation, and all participants received a single infusion of agenT-797 in doses of $100\times$, $300\times$, $1000\times 10^6$ cells. The protocol is included (Supplementary file 1) and appropriate informed consent was obtained from the patient or their duly informed representatives. The study protocol, including the use of patient material, was approved by the institutional review boards at each clinical site. Overall approval was provided by the Saint John's Cancer Institute Clinical Trials Review Committee as part of the Providence Human Research Protection Program.

The primary outcome measures assessed safety (adverse events and dose-limiting toxicities) and secondary measures included: change from baseline in CRP and the number of participants experiencing viral reactivation and fungal infections and the evolution of CRS. Patients were recruited from Weill Cornell Medical College, Norton Cancer Center, and Providence Saint John's Health Center. Patients were enrolled between October 2020 through December 2021. The last trial patient's last visit was June 2022. The EUA patient (IND 29183) was discharged home 1/2023. Last follow up was 7/2023.

### Statistics and reproducibility

Descriptive statistics are used in displaying the results of the primary endpoint: adverse events and dose-limiting toxicities of agent-797 and secondary endpoints related to improvement and resolution of ARDS, evolution of CRS, avoidance of multiorgan dysfunction syndrome, as well as persistence of agent-797. No statistical method was used to predetermine sample size, and no blinding or randomization was performed.

A TEAE is defined as an AE that begins or that worsens in severity after the first dose of the study drug. They are coded using MedDRA version 24.1 with severity assessed according to NCI CTCAE version 5.0. Incidence rates of TEAE are summarized by preferred term (PT) including those attributed to the secondary infections. In addition, Incidence rates of TEAE of grade 3 and above, treatment related TEAE (TRAE), TRAE leading to treatment discontinuation and TRAE leading death are summarized by cohort. For each patient and PT, the worst grade and causality (related to treatment) will be used in the summaries.

Overall survival is defined as the first dose to date of death due to any cause. Kaplan Meier plot of survival is provided to show survival rates at pre-defined timepoints of 30 days and 6 months after the start of therapy. Serum cytokines are grouped into CRS markers and proinflammatory and anti-inflammatory markers and their values are plotted at pre-dose, days 1–7 and days 10–28 post dose. Descriptive $p$-values from ANOVA/mixed models of pre-dose and post-dose serum cytokine level comparisons are provided to show the magnitude of

difference in these exploratory analyses. Plots of WBC, iNKT cells and other pharmacodynamic markers are generated to evaluate the treatment effect on these markers. No data were excluded from analysis.

Due to the small sample size in this exploratory study, results from the statistical tests are not type I error controlled or adequately powered to make any inferential statement, but rather to provide proof of concept for future studies. The statistical analysis plan is included (Supplementary file 2).

### agenT-797 manufacture

agenT-797 is an allogeneic cell therapy product consisting of ≥95% allogeneic human unmodified iNKT cells isolated from one healthy donor mononuclear cell apheresis unit and expanded ex vivo according to cGMP. The drug product is stored frozen. Cells used for agenT-797 are manufactured from mononuclear cell apheresis units from healthy donors homozygous for HLA-A2. Briefly, donor iNKT cells are isolated from apheresis unit by bead-based magnetic separation using monoclonal antibody NKTT120 (proprietary in-house reagent) specific for the iTCR of iNKT cells. Isolated donor iNKT cells undergo two rounds of in vitro stimulation over a 3–4 week period using αGalCer-pulsed irradiated donor peripheral blood mononuclear cells (iPBMCs), after which expanded highly pure iNKT cells (Fig. 1) are harvested and the formulated drug product stored cryopreserved until use. Apheresis donors were recruited from National Marrow Donor Program (NMDP) at 'Be The Match', and Drug product was manufactured under GMP at the Cell Manipulation Core facility/Dana-Farber Cancer Institute (DFCI, Boston MA).

### Cytokine quantification

All supernatants were analyzed using Human Magnetic Luminex Assay (Invitrogen, Th1/Th2 Cytokine 11-Plex Human ProcartaPlex™ Panel, EPX110-10810-901) performed according to the manufacturer's instructions using a FlexMap 3D (Luminex). We used mean fluorescence intensities to analyze differences per analyte between the samples.

In patients, serum biomarkers were measured using MSD-based multiplex assays at the Weill Medical College Clinical and Translational Science Centre Core Lab (New York, NY), and using Myriad's HMP Core 1 and HMP Core 2 multiplex panels at Myriad RBM (Austin, TX).

### Detection of agenT-797 in blood by ddPCR

Detection of agenT-797 in blood was performed using Imegen Quimera technology (Healthincode, Coruna, Spain). Briefly, for each donor/recipient pair informative markers (In/Del) present in donor material (agenT-797) but absent in the recipient was determined by qPCR. Individual informative markers were selected for each donor/patient pair. To detect agenT-797 in treated patients, DNA was extracted from whole blood and analyzed by droplet digital PCR (ddPCR) for the quantity of donor-derived informative markers, and the results expressed as % of agenT-797 in Peripheral Blood Leukocytes.

### Donor specific antibodies

Serum was assessed for the presence of HLA class I and HLA class II panel reactive antibodies (PRA) by flow cytometry (HLA-Antibody detection (PRA) test, Versiti, WI). To determine the HLA-specificity of PRAs, samples positive for the presence of PRAs were further assessed using Luminex-based assays for high resolution HLA antibody determination (HLA Antibody Identification Class I and Class II tests, Versiti, WI). The presence of donor specific antibodies (DSA) was determined by virtual crossmatch of the high-resolution HLA-antibody data with donor HLA-type (Virtual Crossmatch test, Versiti, WI). High resolution donor HLA typing was by sequence-based typing (LabCorp, NC).

### Flow cytometry

Cells were washed with 1xPBS for staining. For Live/Dead staining cells were stained with Zombie NIR Live/Dead Dye (Biolegend) at 1:1000 for

20 min at room temperature and the washed once with FACS Buffer. Fc receptor blocking step was performed using HuTrustain FcX (Biolegend) at 1:100 for 10 min at room temperature protected from the light and then washed once with FACS Buffer. For surface staining, cells were incubated for 30 min at 4 °C with a mix of respective conjugated antibodies. Immediately following staining, cells were once washed with FACS buffer and the samples were analyzed using BD LSR Fortessa. Gating strategies for individual experiments are shown in supplementary Supplementary Fig. 7.

### Co-culture of agenT-797 with dendritic cells and macrophages

Monocytes/Dendritic cells were isolated from cryopreserved PBMCs from healthy HLA-A*02 donors using CD14 microbeads (Miltenyi Biotec, Auburn, CA) and checked for the enrichment using flow-cytometry. For differentiation of monocytes to DC, monocytes were cultured at for 7 days in dendritic cell medium (StemCell) in presence of human GM-CSF, 100 ng/ml (Peprotech, Cranbury, NJ) and human IL-4, 100 ng/ml (Peprotech). After 7 days, DC were added to agenT-797 for co-culture in hTCM for 48 h after which the activation of agenT-797 and DC was assessed using flow-cytometry. For polarization of macrophages, monocytes isolated from cryopreserved PBMCs (StemCell) were polarized either to M1 macrophage using M-CSF at 50 ng/ml, IFN-$\gamma$ at 50 ng/m and LPS at 100 ng/ml or M2 macrophage using M-CSF at 50 ng/ml, IL-10 at 50 ng/ml and TGF-$\beta$1 at 50 ng/ml (Peprotech). After 48-h polarization, M1 and M2 macrophages were labeled with Cell-Tracker Green and co-cultured separately with agenT-797 at 1:2 ratio in hTCM for 48 h after which the activation of agenT-797 as well as viability of M1/M2 macrophages was assessed using flow-cytometry.

### Tumor cell lines

A375-GFP and A375-GFP-CD1d were derived in-house from A375 melanoma cells (*Homo sapiens* malignant melanoma derived female cell line, ATCC: CRL-1619). The cells were cultured in DMEM,1X (Corning, NY) supplemented with 10% FBS, 1% Penicillin-Streptomycin (Gemini, Sacramento, CA). For maintenance of CD1d expression in A375-GFP-CD1d cells, puromycin 1 μg/ml (Gibco) was added during sub-culture. The A375 cell line is HLA-A2 positive and was confirmed by qPCR to express the cancer testes antigen NY-ESO 1.

### Transduction and expansion of NY-ESO TCR$^+$ T cells

T cells were isolated from cryopreserved PBMCs from three healthy HLA-A*02 donors using T cell enrichment kit (StemCell, Vancouver, Canada) following manufacturer' instructions and stimulated using Dynabeads (Gibco, Waltham, MA). After 24 h, NY-ESO-1 lentivirus (A12 Lentivirus, Lentigen, Gaithersburg, MD) was added to T cells at a multiplicity of infection (MOI) of 10 in presence of Lentiboost A and Lentiboost B (Sirion Biotech, Cambridge, MA). After transduction, T cells added to the GRex system (WilsonWolf, Saint Paul, MN) in hTCM media for expansion and were harvested at day 11 after confirming the NYESO-1 TCR expression on CD3+ cells by flow cytometry using a NY-ESO-1 tetramer (MBL International, Waltham, MA). hTCM media was prepared by adding human IL-2, 50 I μ/ml (Roche, Basel, Switzerland) and human IL-7, 2.5 ng/ml (Biolegend, San Diego, CA) to media containing RPMI 1640 Medium (Mod.) 1X with L-Glutamine (Corning, NY), 10% FBS (Gemini, Sacramento, CA), HEPES 10 Mm (Gibco), Penicillin-Streptomycin 50 μ/ml (Gemini, Sacramento, CA), MEM Non-Essential Amino Acids Solution 1X, MEM Amino Acids Solution 1X, B2-mercaptoethanol 50 Mm (Gibco).

### Cytotoxicity and co-culture assays

Cryopreserved antigen exposed NY-ESO TCR$^+$ T Cells and agenT-797 were thawed and rested in hTCM media overnight. For experiments involving agenT-797 supernatant, agenT-797 was added to A375-GFP-CD1d tumor cells (plated overnight) at 1:1 ratio and the supernatant was collected after 24 h of co-culture. Either agenT-797 or supernatant from agenT-797 was added to A375-GFP-CD1d tumor cells (plated overnight)

with antigen exposed T cells at 1:1 ratio in hTCM. The killing of A375-GFP-CD1d cells in co-culture for 5 days was monitored and assessed using Incucyte Live-Cell Analysis System. Killing data presented dynamically as killing curves (% target cell death over time), or as single metric calculated as the definite integral of the killing curve between the start and the end of the experiment (area under the curve; AUC). Additionally, after 24 h of co-culture the supernatant was collected for cytokine analysis using Luminex (Austin, TX) and activation of CD8$^+$ T cells from the co-cultures were assessed using flow-cytometry.

### Repeated antigen exposure of NY-ESO TCR$^+$ T cells

For continuous antigen exposure, A375-GFP cells were irradiated at 8 Gy and NY-ESO TCR$^+$ T Cells were added to the tumor cells at 1:1 ratio 24 h after in hTCM. The cytotoxic capacity of NY-ESO TCR$^+$ T Cells against A375-GFP was determined for each round of antigen exposure. After 48 h of co-culture, supernatant from the co-culture was collected and stored at −80 °C and the number of live T cells present at the end of each round was counted. After completion of each round of antigen exposure, T cells were divided for cryopreservation for phenotyping and continued for consecutive rounds for antigen exposure with fresh tumor cells.

### Reporting summary

Further information on research design is available in the Nature Portfolio Reporting Summary linked to this article.

## Data availability

The authors declare that all data supporting the results in this study are available in the paper and Supplementary Materials. Further data generated in this study are provided in the Supplementary Information. Source data are provided within the paper. Source data are provided with this paper.

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

## Acknowledgements

We are grateful to the patients who were recruited in our study, and their families. agenT-797 used in this study was manufactured by the Connell-O'Reilly Cell Manufacturing Core Facility of the Dana-Farber Cancer Institute, and we are grateful to the support staff there. This study and its funding was supported by MiNK Therapeutics, a subsidiary of Agenus Inc.

## Author contributions

All authors contributed to discussions regarding the data and approved the submitted manuscript. M.A.P. designed and analyzed the clinical monitoring. The versions of the manuscript were written by J.S. and edited by M.A.E., M.A.P., A.B., J.S.B., M.v.D and T.C.H. T.C.H., K.v.B., A.G.A. and D.S. recruited and treated individual patients. W.O. was clinical study leader. J.R., S.N., H.D. and K.S. lead the manufacturing of agenT-797. All other authors specialized in the pre-clinical assays and research and development of agenT-797, collected and analyzed data.

## Competing interests

M.A.P., S.K., D.M., B.Y., S.B., X.M., R.M., A.B., J.S.B., and M.v.D. are current or former employees of MiNK Therapeutics and have received stock and compensation. M.K., A.B., D.C., V.N., W.O., K.S., X.S., J.C., Y.Q. are current employees of Agenus Inc and have received stock and compensation. J.S. from 2020 has been the Editor-in-Chief of Oncogene has sat

on SABs for Vaccitech, Heat Biologics, Eli Lilly, Alveo Technologies, Pear Bio, Agenus (and received compensation), Equilibre Biopharmaceuticals, Graviton Bioscience Corporation, Celltrion, Volvox, Certis, Greenmantle, vTv Therapeutics, APIM Therapeutics, Onconox, IO Labs, Bryologyx, Zephyr AI and Benevolent AI. He has consulted with Lansdowne partners and Vitruvian. He chairs the Board of Directors for Xerion and previously BB Biotech Healthcare Trust PLC. M.A.E. is former employee and current consultant for MiNK, employed by Imvax Inc., I.M. has received consulting fees from Agenus and TCH is on MiNK Therapeutics SAB and a consultant, has received Agenus research funding and consulting fees, and is on the Pfizer International Viral Advisory Council. J.R. received research funding from Equillium, Kite Pharma, Novartis and Oncternal and served on SABs for Akron Biotech, Avrobio, Clade Therapeutics, Erbi Biosystems, Garuda Therapeutics, LifeVault Bio, Novartis, Smart Immune, Talaris Therapeutics and TScan Therapeutics. A.G.A. is on the SAB for Legend Biotech, and a speaker for Bristol Myers Squibb. S.N. sits on the SABs for Kite/Gilead, Iovance, GlaxoSmithKline, Sobi, A2 Bio. K.v.B. has received stock and other ownership from Hemogeny, has a consultant role to HemOgenyx, Glycostem, Gamida Cell, CTI, Intellia, SNIPR BIOME, MorphoSys, Incyte, Autolus, ADC Therapeutics, has received research funding from Precision Biosciences, Orca Bio, Bristol-Myers Squibb/Celgene, Calibr and Actinium Pharmaceuticals. No other author declares a conflict.
