## [Peer Review File · Nature Communications]

A phase 1/2 clinical trial of invariant natural killer T cell therapy in moderate-severe acute respiratory distress syndromeREVIEWER COMMENTS

Reviewer #1 (Remarks to the Author):

Hammond et al has evaluate the therapeutic utility of ex vivo expanded allogeneic iNK T cells (AgenT-797) as novel immune-therapeutics to treat ARDS caused by SARS-2, and reported hat AgenT-797 has been well tolerated without grade 5 event. Correlative studies (Figure 3 -5) showed that recipients have increased level of IL-1Ra (anti-inflammatory agent), but did not have significant changes in both anti- and pro-inflammatory cytokines, and that Agen-T 797 may persist beyond day 21 at higher dose. As a mechanistic studies, authors presented in vitro assay suggesting that AgenT-797 may help enhancing antigen specific conventional T cell functon(CTL) by maintaining activated status (CD69+CD25+) of conventional T cells when co-cultured via unidentified soluble factor (Figure 1), retroactivate antigen presenting dendritic cells and kill (immunosuppressive) M2 macrophages (Figure 2).

Major concern on this work is that none of mechanistic studies presented (Figure 1 and 2) can explain how AgenT-979 help resolving ARDS in setting of severe SARS-2 infection. ARDS from SARS-2 is mostly due to the immune-infiltration and systemic (and local) inflammation from virus. Thus, It is not clear how AgenT-797 help clearing tissue inflammation (or infection). Although authors presented data that AgenT-797 may help rejuvenating antigen specific T cells in vitro, there is no convincing mechanistic data how AgenT-797 help clearing SARS-2 infected cells or virus in tissue (in vivo) or how AgenT-797 suppress tissue inflammation at all. In addition, selective lysis of M2 macrophage by AgenT-797 would support against immunosuppressive effects of AgenT-797.

Followings are minor issues that needs to be clarified

1. Serial CTL Analysis (Figure 1) – better description of how this assay performed would be helpful to understand the results. It is difficult to understand how percent CTL can be decreased after 36-72 hours of co-culture. How long was one round of co-culture? Was target cell irradiated? How was T cell number calculated?
2. Figure 1H-I, What is rationale defining CDD69+CD25+ as activation, rather than CD69+ cells? Was TIGIT assessed on NY-ESO TCR+T cells after co-culture with AgenT-797?
3. Figure 1J – further delineation of soluble factor help maintaining conventional T cell activation
4. Figure 2 showed AgenT-797 selectively lysed M2 – how this could be related to a mechanism of immune regulation by AgenT-797 in setting of SARS2-related ARDS ?
5. Figure 4B. I'm not sure how in vivo distribution of human iNK T cells in xenogenic model can support claims that peripheral persistence of AgenT-797 support tissue localization of AgenT-797 in the lung-disease site.

Reviewer #2 (Remarks to the Author):

This is a Phase 1/2 dose escalation study in which patients with moderate to severe acute respiratory distress syndrome (ARDS) secondary to SARS-CoV-2 or influenza were treated with allogeneic iNKT cells. This is study is important because it is the first study (to my knowledge) in which allogeneic iNKT cells were used as a treatment in virally infected patients. The data suggest that treatment with iNKT cells do not result in overt toxicity, that the adoptively transferred cells persist for at least 6 days, and help to rescue exhausted T cells, target M2 macrophages, and lead to a decrease in pro-inflammatory cytokines in the serum. Moreover, the mouse studies suggest that the iNKT cells can persist the bone marrow and lung, but are rapidly cleared from circulation. However, data are largely correlative, and the data are not well organized.

- 1) Figures 1 and 2 are focused on showing how the authors induce T cell exhaustion using a melanoma model, and then demonstrating showing that the iNKT cell product itself or soluble factors secreted by the iNKT cells able to restore anti-specific T cell responses and that these allo-iNKTs kill M2 macrophages. However, it would be more informative to have a comprehensive characterization of the iNKT cell product (if not RNA-seq data, it would be helpful to know the chemokine receptor profile, if the cells are polyfunctional or skewed towards NKT1/NKT2/NKT17 (Tbet+, GATA+, or RORgt+), as well as if they are polyfunctional, and/or have a Tscm phenotype

- (Toxo+ or Tcf1+). How/ why were these donor iNKT cells selected for use in this study?
- 2) An in vitro tumor model was used to assess the potential therapeutic efficacy of the iNKT cells. The authors could have used an in vitro virus model or assessed correlates of protection in the patient samples. Perhaps iNKT cells are playing a regulatory role and directly decreasing pulmonary inflammation.
- 3) Figures: The figure panels are not individually labeled (A, B, C). Figure 1M is not referred to in legend or text, and the data are hard to analyze. VEGF is misspelled in Fig 3 and it is differences between (right and left panels) VEGF and VEGF-D are unclear.
- 4) Additional references regarding the role of iNKTs in SARS-COV-2 infection could be helpful. Also, there are minor typos throughout the text. For example, p9 line 235 another team "has", methods section p 17 line 462- B2-mecaptoethanol, p 19 propriety in house reagen"t".

Reviewer #3 (Remarks to the Author):

The authors present a phase 1/2 study to evaluate the safety and potential efficacy of agentT-797, an unmodified, allogeneic iNKT cell therapy, in participants with moderate to severe acute respiratory distress syndrome (ARDS) secondary to SARS-CoV-2 or influenza, either with intubation or at high risk to be intubated.

This article presents a potentially interesting study on a clinically important topic related to the treatment of COVID-19. However, there are several concerns regarding the study protocol that raise questions about the interpretation of the data.

1.) The main critique revolves around the absence of a matched control group that did not receive agentT-797 treatment. As discussed by the authors themselves, this absence, along with the variable course of COVID-19, makes it challenging to determine which potential effects can be attributed solely to the therapy and rule out the possibility that they are due to the natural course of the disease.

The supplementary Figure S1, which compares the survival of patients in the study period with CDC data and patients treated outside the study, is of limited help in this context. Firstly, comparisons between "real-life" data and data collected in studies are generally only comparable to a limited extent due to various factors such as differences in patient characteristics, treatment protocols, and data collection methods. Therefore, it is challenging to draw meaningful conclusions from such comparisons. Secondly, the lack of information about the "non-study" patients in Figure S1 makes it impossible to assess whether the groups are truly comparable. Without knowing the characteristics, treatments, and other relevant factors of the "control" patients, it is unclear to what extent the findings can be generalized or attributed to agentT-797 treatment specifically.

2.) Information regarding patient characteristics given in Table 1 are insufficient. In order to properly assess the impact of agentT-797 treatment on the course of COVID-19, it is essential to have a detailed description of patient characteristics and clinical data. These factors can significantly influence the disease progression and treatment outcomes. Important variables that should be more thoroughly described include:

- Pre-existing comorbidities and concomitant medications: The presence of comorbidities and the medications being taken by the patients can have a significant effect on the course of the disease. Thus, it is important to know the specific comorbidities, such as cardiovascular disease, diabetes, or respiratory conditions, as well as the medications being used to manage these conditions. This information would help determine if any observed effects are specific to agentT-797 or if they could be attributed to the pre-existing health conditions or concomitant medications.

- BMI (Body Mass Index): Obesity has been identified as a risk factor for severe COVID-19. Therefore, providing information on the BMI of the patients would be valuable for understanding how this factor might influence the response to AgentT-797 treatment. It would be particularly relevant to know if there were differences in treatment response based on BMI categories.

- Timing of patient enrollment: The timing of patient enrollment in relation to the onset of COVID-

COVID-19 symptoms is crucial for interpreting the results. Patients at different stages of the disease may have different disease trajectories and treatment outcomes. It would be informative to know at what point after the onset of symptoms the patients were included in the study. This information would allow for a better understanding of the potential effects of AgenT-797 at different stages of the disease.

- Previous complications and co-/super-infections: The occurrence of complications before study enrollment is essential to consider when assessing the impact of AgenT-797 treatment. It would be important to know if patients had experienced any significant complications related to COVID-19 before entering the study. Additionally, information on co-infections or superinfections, such as bacterial or fungal infections, at the time of study enrollment would help evaluate the potential effects of these factors on treatment outcomes.

- Other therapeutic measures: Understanding the other therapeutic interventions undertaken before, during, and after study enrollment is crucial. If patients received other treatments for COVID-19, such as glucocorticoids (e.g., dexamethasone) or other immunomodulatory therapies (e.g., tocilizumab, tofacitinib), it is important to document the number of patients receiving these treatments and evaluate their potential impact on the observed outcomes. This information would help assess the specific contribution of AgenT-797 to the observed effects. Providing a comprehensive description of these patient characteristics and clinical data would enhance the understanding of the study's findings and allow for a more accurate assessment of the efficacy and safety of AgenT-797 in the treatment of COVID-19. It would also aid in identifying potential confounding factors and help determine if any observed effects can be attributed solely to the study intervention.

3.) Figure 3 in the article poses challenges for data interpretation due to several limitations, including the absence of an untreated matched control group. Furthermore, the chosen time frame for measurements complicates the assessment of the results.

The analysis includes measurements from day 1 to day 7 after infusion, taken between 2 hours and day 7, as well as measurements from day 10 to day 28, covering a period of 18 days. Given the frequently fluctuating and diverse clinical course observed in patients with severe COVID-19, including complications such as bleeding, coagulation disorders, and superinfections, this extended time frame may not be ideal for accurately detecting the effects of the study agent.

To properly interpret the data, it would be essential to provide information about the complications that occurred at each time point. For example, an increase in IL-8 levels could be attributed to bacterial infections. Including information about complications would enable a more comprehensive understanding of the biomarker data and help assess the potential impact of the study agent.

Additionally, the lack of pre-infusion data for ferritin, CRP, and D-dimers is unclear and prevents an evaluation of the observed changes in these biomarkers over time. Without the baseline data, it becomes impossible to accurately assess the information presented and understand the true impact of the study agent on these biomarkers.

Moreover, neither the figure nor the figure legend provide number of patients analyzed at each time point and whether any differences exist between the study groups. This information is crucial for understanding the statistical power and generalizability of the findings. Without knowing the sample sizes and potential differences between the groups, the interpretation of the results remains limited.

4.) Figure 4A in the article presents data on the proportion of AgenT-797 within the total peripheral blood mononuclear cells (PBMCs). However, the interpretation of these data is challenging due to the lack of information regarding the quantification of PBMCs and the composition of the PBMC pool.

COVID-19 is known to cause significant alterations in the peripheral blood cell counts and composition. These changes can potentially affect the relative proportion of AgenT-797 within the PBMCs. Therefore, it is crucial to provide information on the absolute numbers of PBMCs and the

composition of the PBMC pool to accurately interpret the data presented in Figure 4A.

5.) In the abstract, the authors state that agentT-797 rescued exhausted T cells and rapidly activated both innate and adaptive immunity. However, regarding the rescue of exhausted T cells, only in vitro investigations are shown (Figure 1), and it remains unclear if this can be demonstrated ex vivo in the study patients. Additionally, information about the number of experiments conducted is lacking, which makes it difficult to assess the reliability of the findings. Furthermore, there is no apparent data supporting the rapid activation of innate and/or adaptive immunity. The only evidence presented is in vitro experiments showing the expression of CD80, CD83, CD86, and HLA-DR on MODC (Figure 2). While these findings suggest potential immune cell activation, it remains unclear whether these in vitro observations accurately reflect the in vivo situation.

Answering each reviewers' comments in turn:

Reviewer #1 (Remarks to the Author):

Hammond et al has evaluated the therapeutic utility of ex vivo expanded allogeneic iNK T cells (AgenT-797) as novel immune-therapeutics to treat ARDS caused by SARS-2, and reported that AgenT-797 has been well tolerated without grade 5 event. Correlative studies (Figure 3 -5) showed that recipients have increased level of IL-1Ra (anti-inflammatory agent), but did not have significant changes in both anti- and pro-inflammatory cytokines, and that Agen-T 797 may persist beyond day 21 at higher dose. As a mechanistic studies, authors presented in vitro assay suggesting that AgenT-797 may help enhancing antigen specific conventional T cell function (CTL) by maintaining activated status (CD69+CD25+) of conventional T cells when co-cultured via unidentified soluble factor (Figure 1), retroactivate antigen presenting dendritic cells and kill (immunosuppressive) M2 macrophages (Figure 2).

Major concern on this work is that none of mechanistic studies presented (Figure 1 and 2) can explain how AgenT-979 help resolving ARDS in setting of severe SARS-2 infection. ARDS from SARS-2 is mostly due to the immune-infiltration and systemic (and local) inflammation from virus. Thus, It is not clear how AgenT-797 help clearing tissue inflammation (or infection). Although authors presented data that AgenT-797 may help rejuvenating antigen specific T cells in vitro, there is no convincing mechanistic data how AgenT-797 help clearing SARS-2 infected cells or virus in tissue (in vivo) or how AgenT-797 suppress tissue inflammation at all. In addition, selective lysis of M2 macrophage by AgenT-797 would support against immunosuppressive effects of AgenT-797.

We appreciate the Reviewer's understanding of the trial and concerns they share. We address all of these in the new, improved paper we resubmit. As helpfully suggested, we have extensively enhanced, re-organized and edited the manuscript, changed the order of sections, re-written the majority and added new figures. There are too many examples of this to go through point-by-point and in many ways, this is a completely new paper in response to the comments made.

We include, by way of 1 relevant example, new patient data from a compassionate use case study, which provides independent clinical support for the conclusions reached in the main Covid-19 trial. This was an apparently otherwise healthy young adult male who had cleared SARS-COV2, but was receiving maximal mechanical ventilatory and VV-ECMO for an aspiration pneumonia, which was polymicrobial, including an initially pan-sensitive *Pseudomonas aeruginosa* strain. Despite aggressive treatment, he continued to decline and subsequently developed a carbapenem-resistant *Pseudomonas aeruginosa* pneumonia. With no additional treatment options, we obtained emergency investigational new drug use authorization through the biologics division of the FDA. He quickly stabilized after agenT-797 infusion and fully recovered.

We fully appreciate the limitations here, making efficacy conclusions from non-randomized data and the addition of such a case, although 'the dots begin to connect'. Indeed, just by way of this example, the bronchoalveolar lavage information from this experimental 'one patient' IND (Figure 3), alongside the pre- and post-infusion cytokines from the main trial are very revealing with respect to mechanism (note: addition of this information required addition of 2 new authors, to which all authors agreed).

This information is provided in new figures, but also highlighted in the new methods, results and discussion.

Followings are minor issues that needs to be clarified:

1. Serial CTL Analysis (Figure 1) – better description of how this assay performed would be helpful to understand the results. It is difficult to understand how percent CTL can be decreased after 36-72 hours of co-culture. How long was one round of co-culture? Was target cell irradiated? How was T cell number calculated?

We thank the reviewer for pointing out a lack of some important details, which have now been provided in the text, including the new methods and results with a focus on manufacture.

2. Figure 1H-I, What is rationale defining CDD69+CD25+ as activation, rather than CD69+ cells?

As the reviewer will appreciate, CD69 alone can be a transient activation marker on iNKT, like other T cells. Combined with CD25 they provide a more stringent measure of full activation of iNKT, as described in various publications, such as these (this is now highlighted in the new discussion):

Differential iNKT and T Cells Activation in Non-Alcoholic Fatty Liver Disease and Drug-Induced Liver Injury.

Caballano-Infantes E, García-García A, Lopez-Gomez C, Cueto A, Robles-Diaz M, Ortega-Alonso A, Martín-Reyes F, Alvarez-Alvarez I, Arranz-Salas I, Ruiz-Cabello F, Lucena IM, García-Fuentes E, Andrade RJ, García-Cortes M. Biomedicines. 2021 Dec 28;10(1):55. doi: 10.3390/biomedicines10010055. PMID: 35052736 **Free PMC article.**

Cancer Immunotherapeutic Potential of NKTT320, a Novel, Invariant, Natural Killer T Cell-Activating, Humanized Monoclonal Antibody.

Patel NP, Guan P, Bahal D, Hashem T, Scheuplein F, Schaub R, Nichols KE, Das R. Int J Mol Sci. 2020 Jun 17;21(12):4317. doi: 10.3390/ijms21124317. PMID: 32560408 **Free PMC article.**

Tissue-Specific Phenotype and Activation of iNKT Cells in Morbidly Obese Subjects: Interaction with Adipocytes and Effect of Bariatric Surgery.

López S, García-Serrano S, Gutierrez-Repiso C, Rodríguez-Pacheco F, Ho-Plagaro A, Santiago-Fernandez C, Alba G, Cejudo-Guillen M, Rodríguez-Cañete A, Valdes S, Garrido-Sanchez L, Pozo D, García-Fuentes E. Obes Surg. 2018 Sep;28(9):2774-2782. doi: 10.1007/s11695-018-3215-y. PMID: 29619756

Natural killer T cells constitutively expressing the interleukin-2 receptor \$\alpha\$ chain early in life are primed to respond to lower antigenic stimulation.

Ladd M, Sharma A, Huang Q, Wang AY, Xu L, Genowati I, Levings MK, Lavoie PM. Immunology. 2010 Oct;131(2):289-99. doi: 10.1111/j.1365-2567.2010.03304.x. PMID: 20545784 **Free PMC article**

Invariant NKT cells from HIV-1 or Mycobacterium tuberculosis-infected patients express an activated phenotype.

Montoya CJ, Cataño JC, Ramirez Z, Rugeles MT, Wilson SB, Landay AL. Clin Immunol. 2008 Apr;127(1):1-6. doi: 10.1016/j.clim.2007.12.006. Epub 2008 Mar 4. PMID: 18304877

Was TIGIT assessed on NY-ESO TCR+T cells after co-culture with Agent-797?

TIGIT was not assessed on NY-ESO TCR+ T cells after in vitro agent-797 co-culture however we now add further details of other activation and exhaustion markers in the new manuscript. As R#2 requests as well, despite multiple iTCR stimulation and cytokine driven rapid exponential expansion over a number of weeks, we found little expression of exhaustion / chronic activation markers in the iNKT product, only LAG3 and GITR expression reaching just above 15%, PD-1

approaching 15% and others (TIM3, TIGIT, OX40 and 4-1BB) only between 1 to 7% (new Figure 1D).

3. Figure 1J – further delineation of soluble factor help maintaining conventional T cell activation

While we agree important technically for the further understanding of the *in vitro* studies provided, we believe this substantial new investigation, irrelevant to the conclusions of this clinically-focused study, is beyond the scope of this already extensive manuscript. We now discuss the importance of this point, and a requirement for knockout mouse studies to delineate cytokines/other genes involved, again beyond the scope here.

4. Figure 2 showed AgenT-797 selectively lysed M2 – how this could be related to a mechanism of immune regulation by AgenT-797 in setting of SARS2-related ARDS?

We agree that 'M2' lysis might be expected to bias towards more pro-inflammatory 'Th1' type responses, which can of course be anti-tumor. However, as the Reviewer will well appreciate, M1 and M2 distinctions are not 'black & white' definitions. Indeed, the strongest serological finding of substantial IL-1ra increases shows a different picture (Figure 2a). Moreover, lack of potent impact on CRS cytokines and factors (Figure 2a), as noted by the Reviewer, suggests iNKT cells act beneficially downstream of these, presumably via other (cellular) players and this is where generalizations break down in the complexity *in vivo*, but we have now added new Discussion on this point. We thank the referee for helping provide input into our new, improved paper.

5. Figure 4B. I'm not sure how *in vivo* distribution of human iNK T cells in xenogenic model can support claims that peripheral persistence of AgenT-797 support tissue localization of AgenT-797 in the lung-disease site.

While it would be ideal to have biopsies from many tissues, this is only feasible in models and while xenogenic models are imperfect, we beg to differ on this point. They provide a generally appreciated useful first and preclinical approach, as exploited here in the lung and other organs. Of course, unlike in many more chronic disease settings, we were lucky to only need limited persistence of the iNKT in severe but acute COVID-19, a not unreasonable prospect borne out by the data. We tone down our inferences here in the paper and hope the referee agrees.

Reviewer #2 (Remarks to the Author)

This is a Phase 1/2 dose escalation study in which patients with moderate to severe acute respiratory distress syndrome (ARDS) secondary to SARS-CoV-2 or influenza were treated with allogeneic iNKT cells. This study is important because it is the first study (to my knowledge) in which allogeneic iNKT cells were used as a treatment in virally infected patients. The data suggest that treatment with iNKT cells do not result in overt toxicity, that the adoptively transferred cells persist for at least 6 days, and help to rescue exhausted T cells, target M2 macrophages, and lead to a decrease in pro-inflammatory cytokines in the serum. Moreover, the mouse studies suggest that the iNKT cells can persist the bone marrow and lung, but are rapidly cleared from circulation. However, data are largely correlative, and the data are not well organized.

We appreciate the Reviewer's perceptive understanding of the significance and strengths of the manuscript, limitations inherent in human clinical studies and accept that significant revision was well warranted. We have extensively revised and re-organized the manuscript, now also including new patient data from a compassionate use case study, which provides independent clinical support for the conclusions reached in the main COVID-19 trial.

1) Figures 1 and 2 are focused on showing how the authors induce T cell exhaustion using a melanoma model, and then demonstrating showing that the iNKT cell product itself or soluble factors secreted by the iNKT cells able to restore anti-specific T cell responses and that these allo-iNKTs kill M2 macrophages. However, it would be more informative to have a comprehensive characterization of the iNKT cell product (if not RNA-seq data, it would be helpful to know the chemokine receptor profile, if the cells are polyfunctional or skewed towards NKT1/NKT2/NKT17 (Tbet+, GATA+, or RORgt+), as well as if they are polyfunctional, and/or have a Tscm phenotype (Toxo+ or Tcf1+). How/ why were these donor iNKT cells selected for use in this study?

As the reviewer will appreciate, we focused on extensive characterization of the iNKT product from the perspective of safety for the trial IND. Human T cells as well as iNKT are not always so clearly delineated in subsets as in laboratory mice. Nonetheless, we have shown the high purity, wide range of cytokines (and therefore lack of functional bias) produced by the iNKT product (new Figure 1A, B, C), their cytotoxic activity (e.g. Figure 7) and other features such as ability to stimulate myeloid cells (Figure 6) and reduce conventional T cells exhaustion (e.g. Figure 7). Interestingly and as aforementioned, despite multiple iTCR stimulation and cytokine driven rapid exponential expansion over a number of weeks, we found little expression of exhaustion / chronic activation markers in the iNKT product, only LAG3 and GITR expression reaching just above 15%, PD-1 approaching 15% and others (TIM3, TIGIT, OX40 and 4-1BB) only between 1 to 7% (new Figure 1D).

These particular donors were selected for good health, lack of infections including chronic infections, relatively high iNKT frequency for good yields, lack of cytokine bias common to healthy donor (but not many diseased) iNKT and cytotoxic and other activities. These data are now included.

Furthermore, donors were homozygous for HLA-A02 and linkage disequilibrium favored haplotypes, increasing the potential proportion of recipients in the trial population with at least 2/6 match, in order to obtain persistence information with 0/6 and 2/6 match (at least). More information on these important points has now also been provided in the new manuscript and we are grateful for these comments helping to improve our paper.

2) An *in vitro* tumor model was used to assess the potential therapeutic efficacy of the iNKT cells. The authors could have used an *in vitro* virus model or assessed correlates of protection in the patient samples. Perhaps iNKT cells are playing a regulatory role and directly decreasing pulmonary inflammation.

We agree that these would have been interesting models to pursue. However, cytotoxic assays with acute viral infections *in vitro* suffer from the cytotoxic impact of such viruses (as well as the lack of safety and general availability of specifically COVID-19 *in vitro* models). In general, cell lines (which happen to be tumor cells) are used as uniform targets in such assays.

We thank the Reviewer for the important suggestion that the iNKT could be acting more directly on lung tissue. We have now added this interesting point to the Discussion.

3) Figures: The figure panels are not individually labeled (A, B, C). Figure 1M is not referred to in legend or text, and the data are hard to analyze. VEGF is misspelled in Fig 3 and it is differences between (right and left panels) VEGF and VEGF-D are unclear.

We thank the reviewer for these corrections, which have now been included in the new manuscript. We hope this is better now and apologize for the typos.

4) Additional references regarding the role of iNKTs in SARS-COV-2 infection could be helpful.

This important and relevant point has been included now with supportive references including a new trial of an A2A receptor agonist, Regadenoson, which purports to act via increasing iNKT cells. Studies such as this serve to validate our approach, though we are the first trial of iNKT cells in this setting.

Also, there are minor typos throughout the text. For example, p9 line 235 another team “has”, methods section p 17 line 462- B2-mecaptoethanol, p 19 propriety in house reagent”.

We apologize for these and other errors which we have endeavored to correct. Thank you for spending valuable time pointing out errors such as this, which we should have noticed. We apologize, and are grateful.

Reviewer #3 (Remarks to the Author)

The authors present a phase 1/2 study to evaluate the safety and potential efficacy of agentT-797, an unmodified, allogeneic iNKT cell therapy, in participants with moderate to severe acute respiratory distress syndrome (ARDS) secondary to SARS-CoV-2 or influenza, either with intubation or at high risk to be intubated. This article presents a potentially interesting study on a clinically important topic related to the treatment of COVID-19. However, there are several concerns regarding the study protocol that raise questions about the interpretation of the data.

1.) The main critique revolves around the absence of a matched control group that did not receive agentT-797 treatment. As discussed by the authors themselves, this absence, along with the variable course of COVID-19, makes it challenging to determine which potential effects can be attributed solely to the therapy and rule out the possibility that they are due to the natural course of the disease.

The supplementary Figure S1, which compares the survival of patients in the study period with CDC data and patients treated outside the study, is of limited help in this context. Firstly, comparisons between "real-life" data and data collected in studies are generally only comparable to a limited extent due to various factors such as differences in patient characteristics, treatment protocols, and data collection methods. Therefore, it is challenging to draw meaningful conclusions from such comparisons. Secondly, the lack of information about the “non-study” patients in Figure S1 makes it impossible to assess whether the groups are truly comparable.

Without knowing the characteristics, treatments, and other relevant factors of the “control” patients, it is unclear to what extent the findings can be generalized or attributed to agentT-797 treatment specifically.

We appreciate the Reviewer’s points. We were indeed at pains to point out the limitations of any initial clinical study. However, given the clinical need, ethics obtained, and the importance of first demonstrating safety, we took the necessary approach to achieve that in this first-in-human trial.

We were struck by the results, as the reviewers have noted, and since patients were selected at random among many in the center, the most parsimonious use of this dataset was to make limited comparison with appropriate caveats to those other patients during those depths of the pandemic when treatment options were very limited and therefore most patients (in cohorts two and three) received similar evidence based treatments, including dexamethasone and remdesivir, with the majority also receiving IL-6 antagonists. We have added to the existing tables and provided a new supplementary table with this information in, which we hope suffices now, amongst the new descriptive information in both the methods and results. This is a largely clinical-first paper, and it was important to highlight this. Thank you for pointing this out.

2.) Information regarding patient characteristics given in Table 1 are insufficient. In order to properly assess the impact of agentT-797 treatment on the course of COVID-19, it is essential to have a detailed description of patient characteristics and clinical data. These factors can significantly influence the disease progression and treatment outcomes. Important variables that should be more thoroughly described include:

Pre-existing comorbidities and concomitant medications: The presence of comorbidities and the medications being taken by the patients can have a significant effect on the course of the disease. Thus, it is important to know the specific comorbidities, such as cardiovascular disease, diabetes, or respiratory conditions, as well as the medications being used to manage these conditions. This information would help determine if any observed effects are specific to agentT-797 or if they could be attributed to the pre-existing health conditions or concomitant medications.

We have updated and added the information throughout and apologize for omitting this previously. It is much improved now we hope you agree.

- BMI (Body Mass Index): Obesity has been identified as a risk factor for severe COVID-19. Therefore, providing information on the BMI of the patients would be valuable for understanding how this factor might influence the response to AgentT-797 treatment. It would be particularly relevant to know if there were differences in treatment response based on BMI categories.

We have included information on patient BMI, with higher BMI a known risk factor for severe COVID-19 illness. However, BMI for the treatment groups were not significantly different from comparison cohorts at the same institution, and we point this out. Further, outcomes between different BMI cohorts were not different, by throughout we caution inferring too many conclusions from small datasets.

- Timing of patient enrollment: The timing of patient enrollment in relation to the onset of COVID-19 symptoms is crucial for interpreting the results. Patients at different stages of the disease may have different disease trajectories and treatment outcomes. It would be informative to know at

what point after the onset of symptoms the patients were included in the study. This information would allow for a better understanding of the potential effects of AgenT-797 at different stages of the disease.

All patients enrolled in this early phase study were already intubated and in their second or third week of COVID-19 diagnosis (approximately 18.2 days in Cohort 3 patients), before they were identified, screened, consented and enrolled in this study. Per inclusion criteria, they were intubated and moderate to severe ARDS by Berlin criteria 14 days or less prior to enrollment. Given that many successful interventions for severe COVID-19 (IL-6 antagonists, remdesivir) have demonstrated better efficacy when administered soon after COVID-19 hospitalization, we aim to identify patients in subsequent studies within 72 hours of development of moderate to severe Covid-19 associated acute respiratory distress syndrome (CARDS).

The vast majority of patients were diagnosed with COVID-19 and had been hospitalized for one to three weeks before transfer to the ICU and intubation (range 5-32 days). Consent for iNKT treatment was performed in the ICU from 1-14 days post intubation with all patients noted to have severe ARDS by Berlin Criteria, now referenced. While anecdotal, it was common for patients in the three centers participating in this trial to initially stabilize once hospitalized, then sustain an acute decline in respiratory status in their second week of hospitalization that required higher level of care. This raises the question of whether earlier dosing of agenT-797 might have changed illness trajectory, which can only be answered in well designed, prospective randomized future trials, issues we now address herein.

- Previous complications and co-/super-infections: The occurrence of complications before study enrollment is essential to consider when assessing the impact of AgenT-797 treatment. It would be important to know if patients had experienced any significant complications related to COVID-19 before entering the study.

Most patients (~90%) enrolled had one or more complications of COVID-19 pre-dating agenT-797 infusion. In cohort three, for example, these included antibiotic treated healthcare acquired infections such as *Klebsiella pneumoniae*, *Candida auris* (1 patient), reactivation herpes simplex virus, right ventricular dysfunction by echocardiogram, anemia and thrombocytopenia. However, per the inclusion criteria of this trial, no patients had comorbidities that were expected to limit their survival to < 1 month or preclude safety and efficacy assessment. Patients with significant cardiomyopathy and active systemic bacterial or fungal infections (1 patient returned positive for *C. auris* on a culture from an outside hospital approximately 24 hours after enrollment in the trial) were also excluded as per trial protocol.

These were critically unwell individuals. All patients were extremely sick on enrollment, the VV-ECMO patients in particular. No doubt the reviewer will be aware of the requirements to receive ECMO, including respiratory failure. A major proportion of the patients had various secondary infections, and due this or the risk, all were receiving antibiotics. All secondary infections were treated appropriately with antimicrobials. Thank you for the input here which helps improve the manuscript we resubmit.

Additionally, information on co-infections or superinfections, such as bacterial or fungal infections, at the time of study enrollment would help evaluate the potential effects of these factors on treatment outcomes.

- Other therapeutic measures: Understanding the other therapeutic interventions undertaken before, during, and after study enrollment is crucial. If patients received other treatments for

COVID-19, such as glucocorticoids (e.g., dexamethasone) or other immunomodulatory therapies (e.g., tocilizumab, tofacitinib), it is important to document the number of patients receiving these treatments and evaluate their potential impact on the observed outcomes. This information would help assess the specific contribution of agentT-797 to the observed effects. Providing a comprehensive description of these patient characteristics and clinical data would enhance the understanding of the study's findings and allow for a more accurate assessment of the efficacy and safety of agentT-797 in the treatment of COVID-19. It would also aid in identifying potential confounding factors and help determine if any observed effects can be attributed solely to the study intervention.

We entirely agree with the Reviewer. We would point to existing Table 1 and have now provided more clinical information within the constraints of the journal format and available data. Most of the secondary infections listed in table S1 occurred post infusion of agentT-797. There were 2 urinary tract infections and one lung infection noted on the day of infusion which resolved fully. There were no secondary infections noted before day of infusion. This has been noted in the manuscript. We have included information on concomitant COVID-19 medication (Table 1), which were administered as evidence emerged for efficacy, such as the use of dexamethasone from the Recovery trial. The presence of an active non-COVID-19 infection as per the trial criteria the referee highlights was considered a screen failure and these patients were excluded from the study.

3.) Figure 3 in the article poses challenges for data interpretation due to several limitations, including the absence of an untreated matched control group. Furthermore, the chosen time frame for measurements complicates the assessment of the results.

The analysis includes measurements from day 1 to day 7 after infusion, taken between 2 hours and day 7, as well as measurements from day 10 to day 28, covering a period of 18 days. Given the frequently fluctuating and diverse clinical course observed in patients with severe COVID-19, including complications such as bleeding, coagulation disorders, and superinfections, this extended time frame may not be ideal for accurately detecting the effects of the study agentT.

To properly interpret the data, it would be essential to provide information about the complications that occurred at each time point. For example, an increase in IL-8 levels could be attributed to bacterial infections. Including information about complications would enable a more comprehensive understanding of the biomarker data and help assess the potential impact of the study agentT.

We have updated the information provided including concomitant infections, and infections that occurred post-infusion.

For example, an additional benefit of including data from our eIND patient is that blood samples collected on HD 36 captured an unexpected cytokine response to a secondary inflammation/infection. Clinically, the patient stabilized to a degree post infusion of agentT-797 (HD24) that he was able to undergo a number of procedures in rapid succession, including ECMO oxygenator exchange and percutaneous tracheostomy (HD25) and percutaneous gastrostomy tube (PEG) placement (HD28). On HD 35, he developed a new, single fever to 38.6C and abdominal pain. Imaging studies confirmed accidental dislodgement of his gastrostomy feeding tube and a moderate sized intraperitoneal fluid collection presumed to be extravasated tube feeding. Aside from the fever, which quickly resolved, he remained hemodynamically and clinically stable and was treated with 48 hours of empiric vancomycin and cefepime. Interestingly, blood samples collected on HD 36 demonstrated a significant increase in the cytokines IL-1RA, IL-8

and CXCL10 with a concurrent decrease in IL-12p70, which coincided with identification and bedside removal of his PEG tube. While this observation must be taken in the context of a single, uncontrolled case, it is intriguing that the cytokine profile on HD 36 in this patient demonstrated a robust acute phase and anti-inflammatory response with increased elaboration of CXCL10, potentially as a “homing” signal to CXCR3-positive cells in the presence of inflamed tissue, that coincided with a decrease in the important proinflammatory cytokine IL-12p70. He underwent uncomplicated ECMO decannulation on HD 38.

We have not added much of this information due to space/word limitations but would be very happy to, if you and the Editor agreed (we could include for example a new supplementary section including this individual’s imaging).

Additionally, the lack of pre-infusion data for ferritin, CRP, and D-dimers is unclear and prevents an evaluation of the observed changes in these biomarkers over time. Without the baseline data, it becomes impossible to accurately assess the information presented and understand the true impact of the study agent on these biomarkers.

The day 1 (D1) timepoint for the ferritin, CRP, and D-dimers assessment is pre-infusion, the D1 values thus represent the baseline values. As per clinical protocol, there were no post-infusion samples taken for these markers on Day 1. This has been clarified in both the text and figures, as have the other points raised here. Please accept our apologies for this omission and we thank the referee for helping us to clarify this. We have not laboriously pointed out every change made, but provide a tracked version too so the referees can see the changes made.

Moreover, neither the figure nor the figure legend provide number of patients analyzed at each time point and whether any differences exist between the study groups. This information is crucial for understanding the statistical power and generalizability of the findings. Without knowing the sample sizes and potential differences between the groups, the interpretation of the results remains limited.

These important points are noted and have been addressed in revision. Sample sizes have been clarified in the figure legends. We agree with the reviewer that there are likely differences between these small groups of patients which are potential confounders, but this early phase study establishes safety and sets the stage for randomized, controlled trials for agent-797.

4.) Figure 4A in the article presents data on the proportion of agent-797 within the total peripheral blood mononuclear cells (PBMCs). However, the interpretation of these data is challenging due to the lack of information regarding the quantification of PBMCs and the composition of the PBMC pool. COVID-19 is known to cause significant alterations in the peripheral blood cell counts and composition. These changes can potentially affect the relative proportion of Agent-797 within the PBMCs. Therefore, it is crucial to provide information on the absolute numbers of PBMCs and the composition of the PBMC pool to accurately interpret the data presented in Figure 4A.

We agree with the Reviewer that more information could be provided. We have now provided ALC and normalized % of WBC across the 3 cohorts of the main study (Suppl. Fig. 4). A prior study (ref. 24 in manuscript) has also shown that percentage of iNKT changes reflect absolute numbers in Covid-19 patients. Of course, we have introduced substantial numbers of new healthy donor iNKT cells (approximately equivalent to an individual’s own existing iNKT repertoire).

5.) In the abstract, the authors state that agentT-797 rescued exhausted T cells and rapidly activated both innate and adaptive immunity. However, regarding the rescue of exhausted T cells, only in vitro investigations are shown (Figure 1), and it remains unclear if this can be demonstrated ex vivo in the study patients. Additionally, information about the number of experiments conducted is lacking, which makes it difficult to assess the reliability of the findings. Furthermore, there is no apparent data supporting the rapid activation of innate and/or adaptive immunity. The only evidence presented is in vitro experiments showing the expression of CD80, CD83, CD86, and HLA-DR on MODC (Figure 2). While these findings suggest potential immune cell activation, it remains unclear whether these in vitro observations accurately reflect the in vivo situation.

We accept these limitations and have edited the paper accordingly. As the Reviewer will appreciate, a clinical trial setting does not permit all possible investigations and samples are highly limiting and restricted by consent, which had to be obtained before the results. Thus, not all potential assays were predicted ahead and incorporated in the translational studies and consent forms. The Reviewer will appreciate this particularly in the context of the extreme situation of the most severe part of the challenging and novel SARS-COV2 pandemic. However, because of the generalizable nature of our findings and broad applicability of cellular therapy using iNKT cells, we have changed the title to de-emphasize COVID-19 and hope you agree here. As this reviewer in particular will appreciate, all 21 individuals here were critically unwell, not just the 5 receiving VV-ECMO. We are hugely grateful for the input here, helping us much improve the new paper we resubmit. We hope this referee and the others feel it is now acceptable for publication.

REVIEWER COMMENTS

Reviewer #1 (Remarks to the Author):

This revised version of manuscript is better organized to grasp a clinical impact/endpoint of iNK T cell therapy on patients with respiratory distress, as well as correlative studies. Although I'm not convinced that in vitro mechanistic studies can explain a clinical course, this would be the limitation of any translational studies. Here are minor suggestions
Minor suggestions

Fig 3 Figure legend. Please clarify what "rapid" "secondary" " delayed" mean in the legend. Is it from single donor with triplicate for assay or three donors, with symbol represent a value from donor?

Figure 4. Xenograft data may not be needed in the formal figure. Would consider putting them to supplementary figure

Figure 6C. What would be the cytokine production profile from iNK T cells activated by M1 or M2 microphage? iNK T cells activated by M2 macrophage may be Th2 biased compared to those from M1, and this will be important clue on mechanism of action by iNK T cells in physiologic setting (lung, inflammatory site)

Reviewer #2 (Remarks to the Author):

The authors have significantly revised the manuscript and addressed of my major concerns.

Reviewer #3 (Remarks to the Author):

Firstly, I'd like to acknowledge the evident efforts of the authors in addressing the comments and queries from the reviewers. They have clearly invested significant time and resources in presenting additional data, a dedication that deserves commendation.

However, despite these commendable efforts, my primary concern regarding the study's design remains. The current design of the study yields data that is inherently difficult to interpret, particularly concerning the efficacy of agenT-797.

While the authors have provided additional clinical data, it still presents a challenge to make a robust comparison between the treated patients and the controls. Notably, the two groups show significant differences in clinically relevant parameters. For instance, there's a pronounced difference in the co-medication with established effective therapies. Specifically, the use of dexamethasone is higher in the agenT-797 group (18/20, 80%) compared to the controls (11/20, 55%), with a p-value of 0.02. Additionally, the use of immunomodulatory agents, such as tocilizumab and baricitinib, also shows a marked difference: 60% in the agenT-797 group (12/20) vs. 25% in the controls (5/20), $p=0.03$.

Given these disparities, it remains ambiguous as to whether the variance in mortality between the treated and control groups can be ascribed to the established therapeutic approaches, to the intervention of agenT-797, or if it's merely coincidental.

I regret to note that there remain significant concerns about the interpretability and validity of the study's findings due to its design.

Reviewer #4 (Remarks to the Author):

The authors presented a phase 1/2 study of an unmodified, allogeneic iNKT therapy, in participants with moderate-to-severe ARDS secondary to SARS-CoV-2 or influenza, either with intubation or at high risk to be intubated, to evaluate the safety and potential efficacy of agent-797, an unmodified, allogeneic iNKT cell therapy. They demonstrated that iNKT cells have natural antiviral roles and are known to traffic to the lungs. Their potential for improving both short-term and long-term immunity provide a compelling rationale for their exploitation in SARS-CoV-2 and influenza. There are however several methodological concerns due to insufficient information being provided for the statistical methods/analysis:

- 1) Information on statistical methods is inadequate. The authors mentioned in the protocol (provided as a supplementary document) that statistical analysis plan (SAP) would be a separate document and would include a more technical and detailed description (including templates for tables, listings, and figures) of the planned statistical summaries. However, the SAP was not provided with the submitted article. Hence, information on the implemented statistical approaches to evaluate the potential efficacy of agent-797 is missing and remain unclear in both the report and protocol. Though this is a small exploratory trial, it is still important that essential information for the statistical analyses is provided.
- 2) Definitions of their analysis population sets for their key objectives, e.g., for safety, dose-escalation, and other key endpoints should be reported (in the main paper or appendix). Those were also not found in the protocol.
- 3) Please include a CONSORT flow diagram, which should provide the number of participants assigned to each dose level, received intended treatment, and were analysed for the primary outcome. The losses or exclusions (with reasons) after allocation to each dose level should be clearly presented.
- 4) The authors specified in their protocol that the study used a 3+3 design and incorporated 3 dose levels, with up to 18 patients being needed in the Dose Escalation study component (Part 1) and up to 15 patients in Cohort 4 to further evaluate safety (Part 2, Expansion).
 - They reported: "agent-797 (3 patients at 100 million, 4 at 300 million, 14 at 109 iNKT cells, as a single infusion) was administered to 21 mechanically ventilated patients". The number at each dose level differed for their planned design (which we would expect to be 3 or 6 at the initial doses). Please explain the deviation from the planned 3+3 design.
 - They mentioned that the total sample size was expected to be up to approximately 43 evaluable treated patients. How was the total sample size of 43 determined? (18+15 should be 33) Or did they use a specific test to justify their sample size. If so, please clarify.
- 5) As safety and tolerability were their primary objective, it would be beneficial if the authors can elaborate more on that in their discussion.
- 6) Lastly, the title could be made more informative to include that it is a phase 1/2 trial.

Reviewer #1 (Remarks to the Author):

This revised version of manuscript is better organized to grasp a clinical impact/endpoint of iNK T cell therapy on patients with respiratory distress, as well as correlative studies. Although I'm not convinced that in vitro mechanistic studies can explain a clinical course, this would be the limitation of any translational studies. Here are minor suggestions.

As with other reviewers we enormously appreciate these comments and in general the comments here regarding the manner in which we have updated our paper. Thank you to this referee and the others, and we have addressed all of the comments, which serve to improve the quality of the manuscript we resubmit to you. All files including the figures, supplementary figures and text (including the reporting summary and editorial policy checklist) have been updated, a statistical analysis plan has been included and a tracked version is supplied so that the many changes can be visualized (the figures and changes in these files cannot be 'tracked', as the referees will appreciate).

Minor suggestions

Fig 3 Figure legend. Please clarify what "rapid" "secondary" "delayed" mean in the legend. Is it from single donor with triplicate for assay or three donors, with symbol represent a value from donor?

Data are from the single EUA patient and represent triplicates for assay results. Cytokine response onset classes (rapid, secondary, delayed) have now been appropriately defined in the figure legend and we thank the reviewer for highlighting this.

Figure 4. Xenograft data may not be needed in the formal figure. Would consider putting them to supplementary figure.

We have moved the xenograft data to separate supplementary figure 5, as helpfully suggested.

Figure 6C. What would be the cytokine production profile from iNK T cells activated by M1 or M2 macrophage? iNK T cells activated by M2 macrophage may be Th2 biased compared to those from M1, and this will be important clue on mechanism of action by iNK T cells in physiologic setting (lung, inflammatory site).

We now have a new paragraph in the discussion, with new references to clarify this and think this strengthens the paper (and hope you do too). The iNKT-macrophage interaction is bidirectional, leading to cytokine secretion from both cell types, as demonstrated by the references which we have now added to the manuscript, alongside further discussion.

Specifically, the iNKT interaction with M1 macrophages induces pro-inflammatory cytokine secretion (from both cells), and interaction with M2 macrophages resulting in an anti-inflammatory cytokine signature. In a physiologic setting, the interaction is likely more nuanced, and we can speculate that agent-797 in the inflamed lung modulates the activity of both M1 and M2 macrophages, resulting in the complex changes in lung cytokines observed, as observed in Figure 3. Assuming the changes we observe in lung cytokines derive largely from the iNKT-macrophage interaction, one could argue that agent-797

acts to rapidly reduce secretion of cytokines from both M1 (TNF α , IL-1b) and M2 (IL-10) macrophages, followed by a modified cytokine signature derived from M1 macrophages (IL-6, IL-8, IL-12, IP-10), M2 macrophages (IL-4), CD169+ macrophages (IL-1RA), and iNKT cells (IL-4, IL-5).

Three new, important references elaborating on the bidirectional cellular interaction have been specifically added.

Reviewer #2 (Remarks to the Author):

The authors have significantly revised the manuscript and addressed of my major concerns.

We thank the reviewer for this comment. We really appreciate their support of this study.

Reviewer #3 (Remarks to the Author):

Firstly, I'd like to acknowledge the evident efforts of the authors in addressing the comments and queries from the reviewers. They have clearly invested significant time and resources in presenting additional data, a dedication that deserves commendation.

However, despite these commendable efforts, my primary concern regarding the study's design remains. The current design of the study yields data that is inherently difficult to interpret, particularly concerning the efficacy of agent-797.

While the authors have provided additional clinical data, it still presents a challenge to make a robust comparison between the treated patients and the controls. Notably, the two groups show significant differences in clinically relevant parameters. For instance, there's a pronounced difference in the co-medication with established effective therapies. Specifically, the use of dexamethasone is higher in the agent-797 group (18/20, 80%) compared to the controls (11/20, 55%), with a p-value of 0.02. Additionally, the use of immunomodulatory agents, such as tocilizumab and baricitinib, also shows a marked difference: 60% in the agent-797 group (12/20) vs. 25% in the controls (5/20), p=0.03.

Given these disparities, it remains ambiguous as to whether the variance in mortality between the treated and control groups can be ascribed to the established therapeutic approaches, to the intervention of agent-797, or if it's merely coincidental.

I regret to note that there remain significant concerns about the interpretability and validity of the study's findings due to its design.

We appreciate and understand the Reviewer's concern and appreciate this but would point out that by their very nature, Phase 1/2a single arm studies have substantial validity, but of course cannot usually provide definitive information regarding efficacy. The study could not be powered to show efficacy in the acute setting of a new global pandemic as patients were no longer available after the treated patients had been included. Remarkably we recruited 'the sickest' patients, and there have never been randomized studies of treatments during vv-ECMO (or as this referee will appreciate almost any studies

in this patient group). Thus, although we conclude our abstract stating randomized studies are warranted, these would be very difficult to ever undertake (though we have not in fact discussed that).

Reviewer #4 (Remarks to the Author):

The authors presented a phase 1/2 study of an unmodified, allogeneic iNKT therapy, in participants with moderate-to-severe ARDS secondary to SARS-CoV-2 or influenza, either with intubation or at high risk to be intubated, to evaluate the safety and potential efficacy of agent-797, an unmodified, allogeneic iNKT cell therapy. They demonstrated that iNKT cells have natural antiviral roles and are known to traffic to the lungs. Their potential for improving both short-term and long-term immunity provide a compelling rationale for their exploitation in SARS-CoV-2 and influenza. There are however several methodological concerns due to insufficient information being provided for the statistical methods/analysis:

We understand and appreciate these concerns. We have now included a statistical analysis plan (supplementary protocol file 2). Dr Xin (James) Song, a biostatistician, has now been included as an author here and he has helped clarify the appropriate questions raised below.

1) Information on statistical methods is inadequate. The authors mentioned in the protocol (provided as a supplementary document) that statistical analysis plan (SAP) would be a separate document and would include a more technical and detailed description (including templates for tables, listings, and figures) of the planned statistical summaries. However, the SAP was not provided with the submitted article. Hence, information on the implemented statistical approaches to evaluate the potential efficacy of agent-797 is missing and remain unclear in both the report and protocol. Though this is a small exploratory trial, it is still important that essential information for the statistical analyses is provided.

We agree and provide further information here including the SAP, additional endpoint details, a CONSORT diagram and have updated the text accordingly. The legend of the CONSORT diagram provides additional details, specifically expanding statistical endpoints, and hope that this is now clear.

2) Definitions of their analysis population sets for their key objectives, e.g., for safety, dose-escalation, and other key endpoints should be reported (in the main paper or appendix). Those were also not found in the protocol.

We agree, apologize for this omission and have updated the paper accordingly. We hope that the statistical reviewer is now happy with these changes.

3) Please include a CONSORT flow diagram, which should provide the number of participants assigned to each dose level, received intended treatment, and were analysed for the primary outcome. The losses or exclusions (with reasons) after allocation to each dose level should be clearly presented.

Although CONSORT flow diagrams were designed and intended for randomized clinical trials, we fully agree this is useful information and now present one in the improved and revised manuscript.

4) The authors specified in their protocol that the study used a 3+3 design and incorporated 3 dose levels, with up to 18 patients being needed in the Dose Escalation study component (Part 1) and up to 15 patients in Cohort 4 to further evaluate safety (Part 2, Expansion).

- They reported: “agenT-797 (3 patients at 100 million, 4 at 300 million, 14 at 109 iNKT cells, as a single infusion) was administered to 21 mechanically ventilated patients”. The number at each dose level differed for their planned design (which we would expect to be 3 or 6 at the initial doses). Please explain the deviation from the planned 3+3 design.

The patient cohorts have been clarified in the CONSORT flow diagram and are in line with the planned 3+3 design.

- They mentioned that the total sample size was expected to be up to approximately 43 evaluable treated patients. How was the total sample size of 43 determined? (18+15 should be 33) Or did they use a specific test to justify their sample size. If so, please clarify.

We appreciate the points raised; we stopped recruiting after 20 patients, as the pandemic was subsiding at that point and sites no longer had eligible intubated patients in ICU. The reviewer is correct that the sample size should be 33. It appears that the sample size was incorrectly summed when the protocol was updated from its original (amendment 6, 28 patients) to amendment 8 and we realize this for the first time now. However, this is a post hoc point of no relevance now since we stopped recruiting after 20 patients, as the pandemic was largely over as aforementioned (thanks in part to previous infections and/or vaccines), and sites no longer had eligible intubated patients in ICU.

It is probably worth mentioning that this study, like many, due to the nature of the pandemic at times there was some confusion and disorganization, so our plans did not exactly mirror the SAP: we observed no toxicity so ‘ramped’ up dosing rapidly. We have not provided specific information here but would be happy to, should you and/or the referees so wish and hope you understand the points we raise.

5) As safety and tolerability were their primary objective, it would be beneficial if the authors can elaborate more on that in their discussion.

We hope that the reviewers are in general happier with the updated and improved discussion, and in response to this comment, as the referee may note from the tracked version, we have included new safety comments in the discussion. We have similarly updated both the editorial policy checklist and the reporting summary, accordingly.

6) Lastly, the title could be made more informative to include that it is a phase 1/2 trial.

The title has been updated accordingly; this is again helpful and appropriate. Thank you.

REVIEWERS' COMMENTS

Reviewer #1 (Remarks to the Author):

I appreciate author's all efforts to address reviewers' concern. I think all my critiques were appropriately addressed.

Reviewer #4 (Remarks to the Author):

The authors have satisfactorily addressed my comments. I appreciate the substantial improvements made to the manuscript with sufficient details on the statistical methods as requested, enhancing the overall clarity and rigour of the study.

I am also pleased that the involvement of a statistician in the study has been appropriately acknowledged through co-authorship.